# ON AMORTIZING CONVEX CONJUGATES FOR OPTIMAL TRANSPORT

**Brandon Amos**
Meta AI

## ABSTRACT

This paper focuses on computing the convex conjugate operation that arises when solving Euclidean Wasserstein-2 optimal transport problems. This conjugation, which is also referred to as the Legendre-Fenchel conjugate or $c$-transform, is considered difficult to compute and in practice, Wasserstein-2 methods are limited by not being able to exactly conjugate the dual potentials in continuous space. To overcome this, the computation of the conjugate can be approximated with amortized optimization, which learns a model to predict the conjugate. I show that combining amortized approximations to the conjugate with a solver for fine-tuning significantly improves the quality of transport maps learned for the Wasserstein-2 benchmark by Korotin et al. (2021a) and is able to model many 2-dimensional couplings and flows considered in the literature. All of the baselines, methods, and solvers in this paper are available at http://github.com/facebookresearch/w2ot.

## 1 INTRODUCTION

Optimal transportation (Villani, 2009; Ambrosio, 2003; Santambrogio, 2015; Peyré et al., 2019) is a thriving area of research that provides a way of connecting and transporting between probability measures. While optimal transport between discrete measures is well-understood, e.g. with Sinkhorn distances (Cuturi, 2013), optimal transport between continuous measures is an open research topic actively being investigated (Genevay et al., 2016; Seguy et al., 2017; Taghvaei and Jalali, 2019; Korotin et al., 2019; Makkuva et al., 2020; Fan et al., 2021; Asadulaev et al., 2022). Continuous OT has applications in generative modeling (Arjovsky et al., 2017; Petzka et al., 2017; Wu et al., 2018; Liu et al., 2019; Cao et al., 2019; Leygonie et al., 2019), domain adaptation (Luo et al., 2018; Shen et al., 2018; Xie et al., 2019), barycenter computation (Li et al., 2020; Fan et al., 2020; Korotin et al., 2021b), and biology (Bunne et al., 2021; 2022; Lübeck et al., 2022).

**This paper focuses on estimating the Wasserstein-2 transport map** between measures $\alpha$ and $\beta$ in *Euclidean* space, i.e. $\mathrm{supp}(\alpha) = \mathrm{supp}(\beta) = \mathbb{R}^n$ with the Euclidean distance as the transport cost. The *Wasserstein-2 transport map*, $\check{T} : \mathbb{R}^n \to \mathbb{R}^n$, is the solution to *Monge's primal formulation*:

$$\check{T} \in \operatorname*{arg\,inf}_{T \in \mathcal{T}(\alpha,\beta)} \mathbb{E}_{x \sim \alpha} \|x - T(x)\|_2^2, \tag{1}$$

where $\mathcal{T}(\alpha, \beta) := \{T : T_\# \alpha = \beta\}$ is the set of admissible couplings and the *push-forward operator* $\#$ is defined by $T_\# \alpha(B) := \alpha(T^{-1}(B))$ for a measure $\alpha$, measurable map $T$, and all measurable sets $B$. $\check{T}$ exists and is unique under general settings, e.g. as in Santambrogio (2015, Theorem 1.17), and is often difficult to solve because of the coupling constraints $\mathcal{T}$. **Almost every computational method instead solves the Kantorovich dual**, e.g. as formulated in Villani (2009, §5) and Peyré et al. (2019, §2.5). This paper focuses on the dual associated with the negative inner product cost (Villani, 2009, eq. 5.12), which introduces a *dual potential function* $f : \mathbb{R}^n \to \mathbb{R}$ and solves:

$$\hat{f} \in \operatorname*{arg\,sup}_{f \in L^1(\alpha)} - \mathbb{E}_{x \sim \alpha}[f(x)] - \mathbb{E}_{y \sim \beta}[f^\star(y)] \tag{2}$$

where $L^1(\alpha)$ is the space of measurable functions that are Lebesgue-integrable over $\alpha$ and $f^\star$ is the *convex conjugate*, or *Legendre-Fenchel* transform, of a function $f$ defined by:

$$f^\star(y) := - \inf_{x \in \mathcal{X}} J_f(x; y) \quad \text{with objective} \quad J_f(x; y) := f(x) - \langle x, y \rangle. \tag{3}$$

$\breve{x}(y)$ denotes an optimal solution to eq. (3). Even though the eq. (2) searches over functions in $L^1(\alpha)$, the *optimal dual potential* $\hat{f}$ is convex (Villani, 2009, theorem 5.10). When one of the measures has a density, Brenier (1991, theorem 3.1) and McCann (1995) relate $\hat{f}$ to an optimal transport map $\breve{T}$ for the primal problem in eq. (1) with $\breve{T}(x) = \nabla_x \hat{f}(x)$, and the inverse to the transport map is given by $\breve{T}^{-1}(y) = \nabla_y \hat{f}^{\star}(y)$.

**A stream of foundational papers have proposed methods to approximate the dual potential $f$ with a neural network and learn it by optimizing eq. (2):** Taghvaei and Jalali (2019); Korotin et al. (2019); Makkuva et al. (2020) parameterize $f$ as an *input-convex neural network* (Amos et al., 2017), which can universally represent any convex function with enough capacity (Huang et al., 2020). Other works explore parameterizing $f$ as a *non-convex* neural network (Nhan Dam et al., 2019; Korotin et al., 2021a; Rout et al., 2021).

**Efficiently solving the conjugation operation in eq. (3) is the key computational challenge to solving the Kantorovich dual in eq. (2)** and is an important design choice. Exactly computing the conjugate as done in Taghvaei and Jalali (2019) is considered computationally challenging and approximating it as in Korotin et al. (2019); Makkuva et al. (2020); Nhan Dam et al. (2019); Korotin et al. (2021a); Rout et al. (2021) may be instable. Korotin et al. (2021a) fortifies this observation:

> *The [exact conjugate] solver is slow since each optimization step solves a hard subproblem for computing [the conjugate]. [Solvers that approximate the conjugate] are also hard to optimize: they either diverge from the start or diverge after converging to nearly-optimal saddle point.*

**In contrast to these statements on the difficulty of exactly estimating the conjugate operation, I will show in this paper that computing the (near-)exact conjugate is easy.** My key insight is that the approximate, i.e. *amortized*, conjugation methods can be combined with a fine-tuning procedure using the approximate solution as a starting point. Sect. 3 discusses the amortization design choices and sect. 3.2.2 presents a new amortization perspective on the cycle consistency term used in Wasserstein-2 generative networks (Korotin et al., 2019), which was previously not seen in this way. **Sect. 5 shows that amortizing and fine-tuning the conjugate results in state-of-the-art performance in *all* of the tasks proposed in the Wasserstein-2 benchmark by Korotin et al. (2021a).** Amortization with fine-tuning also nicely models synthetic settings (sect. 6), including for learning a single-block potential flow *without* using the likelihood.

## 2  LEARNING DUAL POTENTIALS: A CONJUGATION PERSPECTIVE

This section reviews the standard methods of learning parameterized dual potentials to solve eq. (2). The first step is to re-cast the Kantorovich dual problem eq. (2) as being over a *parametric* family of potentials $f_\theta$ with parameter $\theta$ as an *input-convex neural network* (Amos et al., 2017) or a more general non-convex neural network. Taghvaei and Jalali (2019); Makkuva et al. (2020) have laid the foundations for optimizing the parametric potentials for the dual objective with

$$\max_\theta \mathcal{V}(\theta) \quad \text{where} \quad \mathcal{V}(\theta) := - \mathbb{E}_{x \sim \alpha} [f_\theta(x)] - \mathbb{E}_{y \sim \beta} [f_\theta^\star(y)] = - \mathbb{E}_{x \sim \alpha} [f_\theta(x)] + \mathbb{E}_{y \sim \beta} [J_{f_\theta}(\breve{x}(y))], \quad (4)$$

where $J$ is the objective to the conjugate optimization problem in eq. (3), $\breve{x}(y)$ is the solution to the convex conjugate, and eq. (4) assumes a finite solution to eq. (2) exists and replaces the sup with a max. Taghvaei and Jalali (2019) show that the model can be *learned*, i.e. the optimal parameters can be found, by taking gradient steps of the dual with respect to the parameters of the potential, i.e. using $\nabla_\theta \mathcal{V}$. This derivative going through the loss and conjugation operation can be obtained by applying *Danskin's envelope theorem* (Danskin, 1966; Bertsekas, 1971) and results in only needing derivatives of the potential:

$$\begin{aligned}
\nabla_\theta \mathcal{V}(\theta) &= \nabla_\theta \left[ - \mathbb{E}_{x \sim \alpha} [f_\theta(x)] + \mathbb{E}_{y \sim \beta} [J_{f_\theta}(\breve{x}(y))] \right] \\
&= - \mathbb{E}_{x \sim \alpha} [\nabla_\theta f_\theta(x)] + \mathbb{E}_{y \sim \beta} [\nabla_\theta f_\theta(\breve{x}(y))]
\end{aligned} \quad (5)$$

where $\breve{x}(y)$ is not differentiated through.

**Assumption 1** *A standard assumption is that the conjugate is smooth with a well-defined* $\arg\min$. *This has been shown to hold when $f$ is strongly convex, e.g. in Kakade et al. (2009), or when $f$ is essentially strictly convex (Rockafellar, 2015, theorem 26.3).*

In practice, assumption 1 in not guaranteed, e.g. non-convex potentials may have a parameterization that results in the conjugate taking infinite values in regions. The dual objective in eq. (2) and eq. (4) discourage the conjugate from diverging as the supremum involves the negation of the conjugate.

**Remark 1** *In eq. (4), the dual potential $f$ associated with the $\alpha$ measure's constraints is the central object that is parameterized and learned, and the dual potential associated with the $\beta$ measure is given by the conjugate $f^\star$ and does not require separately learning. Because of the symmetry of eq. (1), the order can also be **reversed** as in Korotin et al. (2021b) so that the duals associated with the $\beta$ measure are the ones directly parameterized, but we will not consider doing this. Potentials associated with both measures can also be parameterized and we will next see that it is the most natural to think about the model associated with the conjugate as an amortization model.*

**Remark 2** *The dual objective $\mathcal{V}$ can be upper-bounded by replacing $\check{x}$ with any approximation because any sub-optimal solution to the conjugation objective provides an upper-bound to the true objective, i.e. $J(\check{x}(y); y) \leq J(x; y)$ for all $x$. In practice, maximizing a loose upper-bound can cause significant divergence issues as the potential can start over-optimizing the objective.*

Computing the updates to the dual potential's parameters in eq. (5) is a well-defined machine learning setup given a parameterization of the potential $f_\theta$, but is often computationally bottlenecked by the conjugate operation. Because of this bottleneck, many existing work resorts to *amortizing the conjugate* by predicting the solution with a model $\tilde{x}_\phi(y)$. I overview the design choices behind amortizing the conjugate in sect. 3, and then go on in sect. 4 to show that it is reasonable to fine-tune the amortized predictions with an *explicit solver* CONJUGATE$(f, y, x_{\text{init}} = \tilde{x}_\phi(y))$. Algorithm 1 summarizes how to learn a dual potential with an amortized and fine-tuned conjugate.

## 3 AMORTIZING CONVEX CONJUGATES: MODELING AND LOSSES

This section scopes to *predicting* an approximate solution to the conjugate optimization problem in eq. (3). This is an instance of *amortized optimization* methods which predict the solution to a family of optimization problems that are repeatedly solved (Shu, 2017; Chen et al., 2021; Amos, 2022). Amortization is sensible here because the conjugate is repeatedly solved for $y \sim \beta$ every time the dual $\mathcal{V}$ from eq. (4) is evaluated across a batch. Using the basic setup from Amos (2022), I call a prediction to the solution of eq. (3) the *amortization model* $\tilde{x}_\varphi(y)$, which is parameterized by some $\varphi$. The goal is to make the amortization model's prediction match the true conjugate solution, i.e. $\tilde{x}_\phi(y) \approx \check{x}(y)$, for samples $y \sim \beta$. In other words, amortization uses a model to simultaneously solve *all* of the conjugate optimization problems. There are two main design choices: sect. 3.1 discusses *parameterizing the amortization model* and sect. 3.2 overviews *amortization losses*.

### 3.1 PARAMETERIZING A CONJUGATE AMORTIZATION MODEL

The *amortization model* $\tilde{x}_\varphi(y)$ maps a point $y \sim \beta$ to a solution to the conjugate in eq. (3), i.e. $\tilde{x}_\varphi : \mathbb{R}^n \to \mathbb{R}^n$ and the goal is for $\tilde{x}_\varphi(y) \approx \check{x}(y)$. In this paper, I take standard potential models further described in app. B and keep them fixed to ablate across the amortization loss and fine-tuning choices. The main categories are:

1. $\tilde{x}_\varphi : \mathbb{R}^n \to \mathbb{R}^n$ *directly maps to the solution* of eq. (3) with a multilayer perceptron (MLP) as in Nhan Dam et al. (2019), or a U-Net (Ronneberger et al., 2015) for image-based transport. These are also used in parts of Korotin et al. (2021a).

2. $\tilde{x}_\varphi = \nabla_y g_\varphi$ is the *gradient of a function* $g_\varphi : \mathbb{R}^n \to \mathbb{R}$. Korotin et al. (2019); Makkuva et al. (2020) parameterize $g_\varphi$ as an input-convex neural network, and some methods of Korotin et al. (2021a) parametrize $g_\varphi$ as a ResNet (He et al., 2016). This is well-motivated because the $\arg\min$ of a convex conjugate is the derivative, i.e. $\check{x}(y) = \nabla_y f^\star(y)$.

---

**Algorithm 1** Learning Wasserstein-2 dual potentials with amortized and fine-tuned conjugation

---

**Inputs:** Measures $\alpha$ and $\beta$ to couple, initial dual potential $f_\theta$, and initial amortization model $\tilde{x}_\varphi$
**while** unconverged **do**
    *Sample batches* $\{x_j\} \sim \alpha$ and $\{y_j\} \sim \beta$ indexed by $j \in [N]$
    Obtain the *amortized prediction* of the conjugate $\tilde{x}_\varphi(y_j)$
    *Fine-tune the prediction* by numerically solving $\check{x}(y_j) = \text{CONJUGATE}(f, y_j, x_{\text{init}} = \tilde{x}_\varphi(y_j))$
    *Update the potential* with a gradient estimate of the dual in eq. (5), i.e. $\nabla_\theta \mathcal{V}$
    *Update the amortization model* with a gradient estimate of a loss from sect. 3, i.e. $\nabla_\varphi \mathcal{L}$
**end while**
**return** optimal dual potentials $f_\theta$ and conjugate amortization model $\tilde{x}_\varphi$

---

### 3.2 CONJUGATE AMORTIZATION LOSS CHOICES

We now turn to the design choice of what loss to optimize so that the conjugate amortization model $\tilde{x}_\varphi$ best-predicts the solution to the conjugate. In all cases, the loss is differentiable and $\varphi$ is optimized with a gradient-based optimizer. I present an amortization perspective of methods not previously presented as amortization methods, which is useful to help think about improving the amortized predictions with the fine-tuning and exact solvers in sect. 4. Figure 1 illustrates the main loss choices.

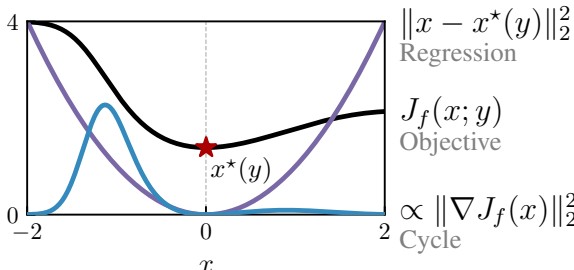

Figure 1: Conjugate amortization losses.

#### 3.2.1 OBJECTIVE-BASED AMORTIZATION

Nhan Dam et al. (2019) propose to make the amortized prediction optimal on the *conjugation objective* $J_f$ from eq. (3) across samples from $\beta$, i.e.:

$$\min_\varphi \mathcal{L}_{\text{obj}}(\varphi) \text{ where } \mathcal{L}_{\text{obj}}(\varphi) := \mathbb{E}_{y \sim \beta} J_f(\tilde{x}_\varphi(y); y). \tag{6}$$

We refer to $\mathcal{L}_{\text{obj}}$ as *objective-based* amortization and solve eq. (6) by taking gradient steps $\nabla_\varphi \mathcal{L}_{\text{obj}}$ using a Monte-Carlo estimate of the expectation.

**Remark 3** *The maximin method proposed in Makkuva et al. (2020, theorem 3.3) is equivalent to maximizing an upper-bound to the dual loss $\mathcal{V}$ with respect to $\theta$ of a potential $f_\theta$ and minimizing the objective-based amortization loss $\mathcal{L}_{\text{obj}}$ with respect to $\varphi$ of an amortization model $\tilde{x}_\varphi := \nabla g_\varphi$. Their formulation replaces the exact conjugate $\check{x}$ in eq. (4) with an approximation $\tilde{x}_\varphi$, i.e.:*

$$\max_\theta \min_\varphi \mathcal{V}_{\text{MM}}(\theta, \varphi) \text{ where } \mathcal{V}_{\text{MM}}(\theta, \varphi) := -\mathbb{E}_{x \sim \alpha}[f_\theta(x)] + \mathbb{E}_{y \sim \beta}[J_{f_\theta}(\tilde{x}_\varphi(y); y)]. \tag{7}$$

*Makkuva et al. (2020) propose to optimize $\mathcal{V}_{\text{MM}}$ with gradient ascent-descent steps. For optimizing $\theta$, $\mathcal{V}_{\text{MM}}(\theta, \varphi)$ is an upper bound on the true dual objective $\mathcal{V}(\theta)$ as discussed in remark 2 with equality if and only if $\tilde{x}_\varphi = \check{x}$.* **Evaluating the inner optimization step is exactly the objective-based amortization update,** *i.e.,* $\nabla_\varphi \mathcal{V}_{\text{MM}}(\theta, \varphi) = \nabla_\varphi \mathcal{L}_{\text{obj}}(\varphi) = \nabla_\varphi J_{f_\theta}(\tilde{x}_\varphi(y); y)$.

**Remark 4** *Suboptimal predictions of the conjugate often leads to a divergent upper bound on $\mathcal{V}(\theta)$. Makkuva et al. (2020, algorithm 1) propose to fix this by running more updates on the amortization model. In sect. 4, I propose fine-tuning as an alternative to obtain a near-exact conjugates.*

#### 3.2.2 FIRST-ORDER OPTIMALITY AMORTIZATION: CYCLE CONSISTENCY AND W2GN

An alternative to optimizing the dual objective directly as in eq. (6) is to optimize for the *first-order optimality condition*. Eq. (3) is an unconstrained minimization problem, so the first-order optimality condition is that the derivative of the objective is zero, i.e. $\nabla_x J_f(x; y) = \nabla_x f(x) - y = 0$. The conjugate amortization model can be optimized for the *residual norm* of this condition with

$$\min_\varphi \mathcal{L}_{\text{cycle}}(\varphi) \text{ where } \mathcal{L}_{\text{cycle}}(\varphi) := \mathbb{E}_{y \sim \beta} \|\nabla_x J_f(\tilde{x}_\varphi(y); y)\|_2^2 = \mathbb{E}_{y \sim \beta} \|\nabla_x f(\tilde{x}_\varphi(y)) - y\|_2^2. \tag{8}$$

**Remark 5** *W2GN (Korotin et al., 2019) is equivalent to maximizing an upper-bound to the dual loss $\mathcal{V}$ with respect to $\theta$ of a potential $f_\theta$ and minimizing the first-order amortization loss $\mathcal{L}_{\text{cycle}}$ with respect to $\varphi$ of an conjugate amortization model $\tilde{x}_\varphi := \nabla g_\varphi$. Korotin et al. (2019) originally motivated the cycle consistency term from the use in cross-domain generative modeling Zhu et al. (2017) and **eq. (8) shows an alternative way of deriving the cycle consistency term by amortizing the first-order optimality conditions of the conjugate.***

**Remark 6** *The formulation in Korotin et al. (2019) does **not** disconnect $f_\theta$ when optimizing the cycle loss in eq. (8). From an amortization perspective, this performs amortization by updating $f_\theta$ to have a solution closer to $\tilde{x}_\varphi$ rather than the usual amortization setting of updating $\tilde{x}_\varphi$ to make a prediction closer to the solution of $f_\theta$. In my experiments, updating $f_\theta$ with the amortization term seems to help when not fine-tuning the conjugate to be exact, but not when using the exact conjugates.*

**Remark 7** *Korotin et al. (2019) and followup papers such as Korotin et al. (2021b) state that they do not perform maximin optimization as in eq. (7) from Makkuva et al. (2020) because they replace the inner optimization of the conjugate with an approximation. **I disagree that the main distinction between these methods should be based on their formulation as a maximin optimization problem.** I instead propose that the main difference between their losses is how they amortize the convex conjugate: Makkuva et al. (2020) use the objective-based loss in eq. (6) while Korotin et al. (2019) use the first-order optimality condition (eq. (8)). Sect. 5 shows that adding fine-tuning and exact conjugates to both of these methods makes their performance match in most cases.*

**Remark 8** *Optimizing for the first-order optimality conditions may not be ideal for non-convex conjugate objectives as inflection points with a near-zero derivative may not be a global minimum of eq. (3). The left and right regions of fig. 1 illustrate this.*

### 3.2.3 REGRESSION-BASED AMORTIZATION

The previous objective and first-order amortization methods locally refine the model's prediction using local derivative information. The conjugate amortization model can also be trained by regressing onto ground-truth solutions when they are available, i.e.

$$\min_\varphi \mathcal{L}_{\text{reg}}(\varphi) \text{ where } \mathcal{L}_{\text{reg}}(\varphi) := \mathop{\mathbb{E}}_{y \sim \beta} \|\tilde{x}_\varphi(y) - \breve{x}(y)\|_2^2. \tag{9}$$

This regression loss is the most useful when approximations to the conjugate are computationally easy to obtain, e.g. with a method described in sect. 4. $\mathcal{L}_{\text{reg}}$ gives the amortization model information about where the globally optimal solution is rather than requiring it to only locally search over the conjugate's objective $J$.

## 4 NUMERICAL SOLVERS FOR EXACT CONJUGATES AND FINE TUNING

In the Euclidean Wasserstein-2 setting, the conjugation operation in eq. (3) is a continuous and unconstrained optimization problem over a possibly non-convex potential $f$. It is usually implemented with a method using first-order information for the update in algorithm 2, such as:

---
**Algorithm 2** CONJUGATE($f, y, x_{\text{init}}$)

---
$x \leftarrow x_{\text{init}}$
**while** unconverged **do**
    Update $x$ with $\nabla_x J_f(x; y)$
**end while**
**return** optimal $\breve{x}(y) = x$

---

1. *Adam* (Kingma and Ba, 2014) is an adaptive first-order optimizer for high-dimensional optimization problems and is used for the exact conjugations in Korotin et al. (2021a). **Note:** Adam here is for algorithm 2 and is *not* performing parameter optimization.

2. *L-BFGS* (Liu and Nocedal, 1989) is a quasi-Newton method for optimizing unconstrained convex functions. App. A discusses more implementation details behind setting up L-BFGS efficiently to run on the batches of optimization problems considered here. Choosing the line search method is the most crucial part as the conditional nature of some line searches may be prohibitive over batches. Table 3 shows that an Armijo search often works well to obtain approximate solutions.

## 5 EXPERIMENTAL RESULTS ON THE WASSERSTEIN-2 BENCHMARK

I have focused most of the experimental investigations on the Wasserstein-2 benchmark (Korotin et al., 2021a) because it provides a concrete evaluation setting with established baselines for learning potentials for Euclidean Wasserstein-2 optimal transport. The tasks in the benchmark have known (ground-truth) optimal transport maps and include transporting between: 1) high-dimensional (HD) mixtures of Gaussians, and 2) samples from generative models trained on CelebA (Liu et al., 2015). The main evaluation metric is the *unexplained variance percentage* ($\mathcal{L}^2$-UVP) metric from (Korotin et al., 2019), which compares a candidate map $T$ to the ground truth map $T^\star$ with:

$$\mathcal{L}^2\text{-UVP}(T; \alpha, \beta) := 100 \cdot \|T - T^\star\|_{\mathcal{L}^2(\alpha)}^2 / \text{Var}(\beta)\%. \tag{10}$$

In all of the experimental results, I report the *final $\mathcal{L}^2$-UVP* evaluated with 16384 samples at the end of training, and average the results over 10 trials. App. C further details the experimental setup. My original motivation for running these experiments was to understand how ablating the amortization losses and fine-tuning options impacts the final $\mathcal{L}^2$-UVP performance of the learned potential.

**The main experimental takeaway of this paper is that fine-tuning the amortized conjugate with a solver significantly improves the learned transport maps.** Tables 1 and 2 report that amortizing and fine-tuning the conjugate improves the $\mathcal{L}^2$-UVP performance by a factor of 1.8 to 4.4 over the previously best-known results on the benchmark. App. C.3 shows that the conjugate can often be fine-tuned within 100ms per batch of 1024 examples on an NVIDIA Tesla V100 GPU, fig. 2 and app. C.2 compare Adam and L-BFGS for solving the conjugation. The following remarks further summarize the results from these experiments:

**Remark 9** *With fine-tuning, the choice of regression or objective-based amortization doesn't significantly impact the $\mathcal{L}^2$-UVP of the final potential. This is because fine-tuning is usually able to find the optimal conjugates from the predicted starting points.*

**Remark 10** *My re-implementation of W2GN (Korotin et al., 2019), which uses cycle consistency amortization with no fine-tuning, often outperforms the results reported in Korotin et al. (2021a). This is likely due to differences in the base potential and conjugate amortization models.*

**Remark 11** *Cycle consistency sometimes provides difficult starting points for the fine-tuning methods, especially for L-BFGS. When learning non-convex potentials, this poor performance is likely related to the fact that **Newton methods are known to be difficult for saddle points** (Dauphin et al., 2014). Combining cycle consistency, which tries to find a point where the derivative is zero, with L-BFGS, which also tries to find a point where the derivative is zero, results in finding suboptimal inflection points of the potential rather than the true minimizer.*

**Remark 12** *The performance of the methods using objective-based amortization without fine-tuning, as done in Taghvaei and Jalali (2019), are lower than the performance reported in Korotin et al. (2021a). This is because I do not run multiple inner updates to update the conjugate amortization model. I instead advocate for fine-tuning the conjugate predictions with a known solver, eliminating the need for a hyper-parameter of the number of inner iterations that needs to be delicately tuned to make sure the amortized prediction alone does not diverge too much from the true conjugate.*

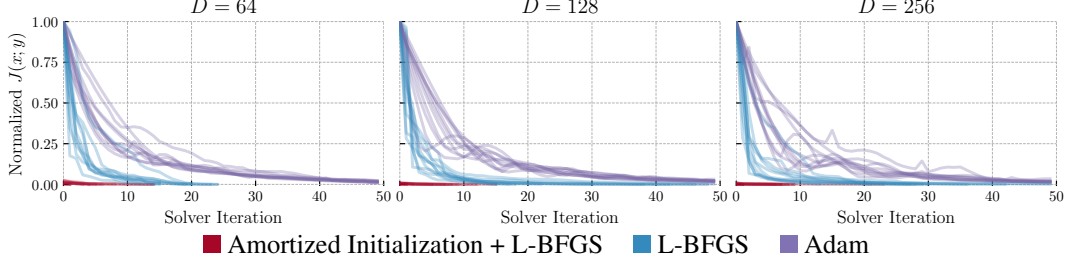

Figure 2: Conjugate solver convergence on the HD benchmarks with an ICNN potential.

Table 1: Comparison of $\mathcal{L}^2$-UVP on the high-dimensional tasks from the Wasserstein-2 benchmark by Korotin et al. (2021a), where *[the gray tags] denote their results. I report the mean and standard deviation across 10 trials. **Fine-tuning the amortized prediction with L-BFGS or Adam consistently improves the quality of the learned potential.**

**Baselines** from Korotin et al. (2021a)

| | Amortization loss | Conjugate solver | $n=2$ | $n=4$ | $n=8$ | $n=16$ | $n=32$ | $n=64$ | $n=128$ | $n=256$ |
|---|---|---|---|---|---|---|---|---|---|---|
| *[W2] | Cycle | None | 0.1 | 0.7 | 2.6 | 3.3 | 6.0 | 7.2 | 2.0 | 2.7 |
| *[MMv1] | None | Adam | 0.2 | 1.0 | 1.8 | 1.4 | 6.9 | 8.1 | 2.2 | 2.6 |
| *[MMv2] | Objective | None | 0.1 | 0.68 | 2.2 | 3.1 | 5.3 | 10.1 | 3.2 | 2.7 |
| *[MM] | Objective | None | 0.1 | 0.3 | 0.9 | 2.2 | 4.2 | 3.2 | 3.1 | 4.1 |

**Potential model:** the input convex neural network described in app. B.3      **Amortization model:** the MLP described in app. B.2

| Amortization loss | Conjugate solver | $n=2$ | $n=4$ | $n=8$ | $n=16$ | $n=32$ | $n=64$ | $n=128$ | $n=256$ |
|---|---|---|---|---|---|---|---|---|---|
| Cycle | None | 0.28 ±0.09 | 0.90 ±0.11 | 2.23 ±0.20 | 3.03 ±0.06 | 5.32 ±0.14 | 8.79 ±0.16 | 5.66 ±0.45 | 4.34 ±0.14 |
| Objective | None | 0.27 ±0.09 | 0.78 ±0.12 | 1.78 ±0.26 | 2.00 ±0.11 | >100 | >100 | >100 | >100 |
| Cycle | L-BFGS | 0.26 ±0.09 | 0.77 ±0.11 | 1.63 ±0.28 | 1.15 ±0.14 | 2.02 ±0.10 | 4.48 ±0.89 | 1.65 ±0.10 | 5.93 ±9.43 |
| Objective | L-BFGS | 0.26 ±0.09 | 0.79 ±0.12 | 1.63 ±0.30 | 1.12 ±0.11 | 1.92 ±0.19 | 4.40 ±0.79 | 1.64 ±0.11 | 2.24 ±0.13 |
| Regression | L-BFGS | 0.26 ±0.09 | 0.78 ±0.12 | 1.64 ±0.29 | 1.14 ±0.12 | 1.93 ±0.20 | 4.41 ±0.74 | 1.69 ±0.11 | 2.21 ±0.15 |
| Cycle | Adam | 0.26 ±0.09 | 0.79 ±0.11 | 1.62 ±0.29 | 1.14 ±0.12 | 1.95 ±0.21 | 4.55 ±0.62 | 1.88 ±0.26 | >100 |
| Objective | Adam | 0.26 ±0.09 | 0.79 ±0.14 | 1.62 ±0.31 | 1.08 ±0.14 | 1.89 ±0.19 | 4.23 ±0.76 | 1.59 ±0.12 | 1.99 ±0.15 |
| Regression | Adam | 0.35 ±0.07 | 0.81 ±0.12 | 1.61 ±0.32 | 1.09 ±0.11 | 1.85 ±0.20 | 4.42 ±0.68 | 1.63 ±0.08 | 1.99 ±0.16 |

**Potential model:** the non-convex neural network (MLP) described in app. B.4      **Amortization model:** the MLP described in app. B.2

| Amortization loss | Conjugate solver | $n=2$ | $n=4$ | $n=8$ | $n=16$ | $n=32$ | $n=64$ | $n=128$ | $n=256$ |
|---|---|---|---|---|---|---|---|---|---|
| Cycle | None | 0.05 ±0.00 | 0.35 ±0.01 | 1.51 ±0.08 | >100 | >100 | >100 | >100 | >100 |
| Objective | None | >100 | >100 | >100 | >100 | >100 | >100 | >100 | >100 |
| Cycle | L-BFGS | >100 | >100 | >100 | >100 | >100 | >100 | >100 | >100 |
| Objective | L-BFGS | 0.03 ±0.00 | 0.22 ±0.01 | 0.60 ±0.03 | 0.80 ±0.11 | 2.09 ±0.31 | 2.08 ±0.40 | 0.67 ±0.05 | 0.59 ±0.04 |
| Regression | L-BFGS | 0.03 ±0.00 | 0.22 ±0.01 | 0.61 ±0.04 | 0.77 ±0.10 | 1.97 ±0.38 | 2.08 ±0.39 | 0.67 ±0.05 | 0.65 ±0.07 |
| Cycle | Adam | 0.18 ±0.03 | 0.69 ±0.56 | 1.62 ±2.82 | >100 | >100 | >100 | >100 | >100 |
| Objective | Adam | 0.06 ±0.01 | 0.26 ±0.02 | 0.63 ±0.07 | 0.81 ±0.10 | 1.99 ±0.32 | 2.21 ±0.32 | 0.77 ±0.05 | 0.66 ±0.07 |
| Regression | Adam | 0.22 ±0.01 | 0.28 ±0.02 | 0.61 ±0.07 | 0.80 ±0.10 | 2.07 ±0.38 | 2.37 ±0.46 | 0.77 ±0.06 | 0.75 ±0.09 |
| Improvement factor over prior work | | **3.3** | **3.1** | **3.0** | **1.8** | **2.7** | **1.5** | **3.0** | **4.4** |

Table 2: Comparison of $\mathcal{L}^2$-UVP on the CelebA64 tasks from the Wasserstein-2 benchmark by Korotin et al. (2021a), where *[the gray tags] denote their results. I report the mean and standard deviation across 10 trials. **Fine-tuning the amortized prediction with L-BFGS or Adam consistently improves the quality of the learned potential.** The ConvICNN64 and ResNet potential models are from Korotin et al. (2021a), and app. B.5 describes the (non-convex) ConvNet model.

| | Amortization loss | Conjugate solver | Potential Model | Early Generator | Mid Generator | Late Generator |
|---|---|---|---|---|---|---|
| *[W2] | Cycle | None | ConvICNN64 | 1.7 | 0.5 | 0.25 |
| *[MM] | Objective | None | ResNet | 2.2 | 0.9 | 0.53 |
| *[MM-R†] | Objective | None | ResNet | 1.4 | 0.4 | 0.22 |
| | Cycle | None | ConvNet | >100 | 26.50 ±60.14 | 0.29 ±0.59 |
| | Objective | None | ConvNet | >100 | 0.29 ±0.15 | 0.69 ±0.90 |
| | Cycle | Adam | ConvNet | 0.65 ±0.02 | 0.21 ±0.00 | 0.11 ±0.04 |
| | Cycle | L-BFGS | ConvNet | 0.62 ±0.01 | 0.20 ±0.00 | 0.09 ±0.00 |
| | Objective | Adam | ConvNet | 0.65 ±0.02 | 0.21 ±0.00 | 0.11 ±0.05 |
| | Objective | L-BFGS | ConvNet | 0.61 ±0.01 | 0.20 ±0.00 | 0.09 ±0.00 |
| | Regression | Adam | ConvNet | 0.66 ±0.01 | 0.21 ±0.00 | 0.12 ±0.00 |
| | Regression | L-BFGS | ConvNet | 0.62 ±0.01 | 0.20 ±0.00 | 0.09 ±0.01 |
| | Improvement factor over prior work | | | **2.3** | **2.0** | **2.4** |

†the *reversed* direction from Korotin et al. (2021a), i.e. the potential model is associated with the $\beta$ measure

## 6 DEMONSTRATIONS ON SYNTHETIC DATA

I lastly demonstrate the stability of amortization and fine-tuning as described in algorithm 1 to learn optimal transport maps between many 2d synthetic settings considered in the literature. In all of these settings, I instantiate ICNN and MLP architectures and use regression-based amortization with L-BFGS fine-tuning. Figures 3 to 5 show the settings considered in Makkuva et al. (2020) and Rout et al. (2021), and fig. 6 shows the conjugate objective landscapes. Figure 7 shows maps learned on synthetic settings from Huang et al. (2020). App. D contains more experimental details here.

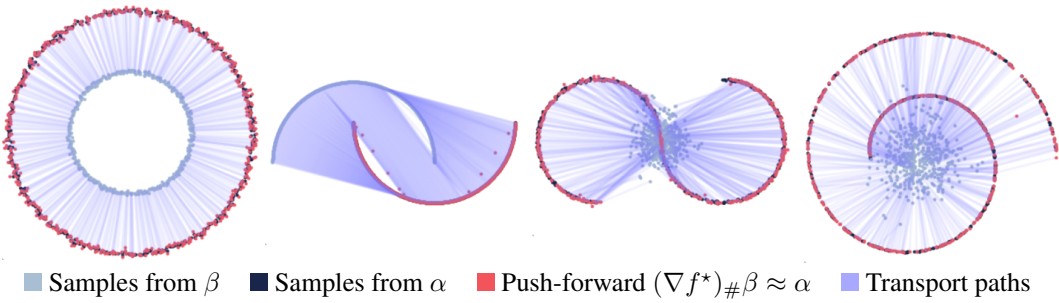

■ Samples from $\beta$   ■ Samples from $\alpha$   ■ Push-forward $(\nabla f^\star)_{\#}\beta \approx \alpha$   ■ Transport paths

Figure 3: Learned transport maps on synthetic settings from Rout et al. (2021).

**Remark 13** *Optimizing the dual in eq. (4) is an alternative to the maximum likelihood training in Huang et al. (2020) for potential flows. While maximum likelihood training requires the density of one of the measures, optimizing the dual only requires samples from the measures. This makes it easy to compute flows such as the bottom one of fig. 7, even though it is difficult to specify the density.*

## 7 RELATED WORK

**Numerical conjugation.** Brenier (1989); Lucet (1996; 1997); Gardiner and Lucet (2013); Trienis (2007); Jacobs and Léger (2020); Vacher and Vialard (2021) also use computational methods for numerical conjugation, which also have applications in solving Hamilton-Jacobi or Burger's equation via discretizations. These methods typically consider conjugating univariate and bivariate functions by discretizing the space, which make them challenging to apply in the settings from Korotin et al. (2021a) that we report in sect. 5: we are able to conjugate dual potentials in up to 256-dimensional spaces for the HD tasks and 12228-dimensional spaces for the CelebA64 tasks. Taking a grid-based discretization of the space with 10 locations in each dimension would result in $(12228)^{10}$ grid points in the CelebA64 task. Garcia et al. (2023) amortizes the conjugate operation to predict the natural gradient update, which is related to the amortized proximal optimization setting in Bae et al. (2022).

**Learning solutions to OT problems.** Dinitz et al. (2021); Khodak et al. (2022); Amos et al. (2022) amortize and learn the solutions to OT and matching problems by predicting the optimal duals given the input measures. These approaches are complimentary to this paper as they amortize the *solution* to the dual in eq. (2) while this paper amortizes the conjugate subproblem in eq. (3) that is repeatedly computed when solving a single OT problem.

## 8 CONCLUSIONS, FUTURE DIRECTIONS, AND LIMITATIONS

This paper explores the use of amortization and fine-tuning for computing convex conjugates. The methodological insights and amortization perspective may directly transfer to many other applications and extensions of Euclidean Wasserstein-2 optimal transport, including for computing barycenters (Korotin et al., 2021b), Wasserstein gradient flows (Alvarez-Melis et al., 2021; Mokrov et al., 2021), or cellular trajectories (Bunne et al., 2021). Many of the key amortization and fine-tuning concepts from here will transfer beyond the Euclidean Wasserstein-2 setting, e.g. the more general $c$-transform arising in non-Euclidean optimal transport (Sei, 2013; Cohen et al., 2021; Rezende and Racanière, 2021) or for the Moreau envelope computation, which can be decomposed into a term that involve the convex conjugate as described in Rockafellar and Wets (2009, ex. 11.26) and Lucet (2006, sect. 2).

**Limitations.** The most significant limitation in the field of estimating Euclidean Wasserstein-2 optimal transport maps is the lack of convergence guarantees. The parameter optimization problem in eq. (4) is *always* non-convex, even when using input-convex neural networks. I have shown that improved conjugate estimations significantly improve the stability when the base potential model is properly set up, but **all methods are sensitive to the potential model's hyper-parameters.** I found that small changes to the activation type or initial learning rate can cause *no* method to converge.

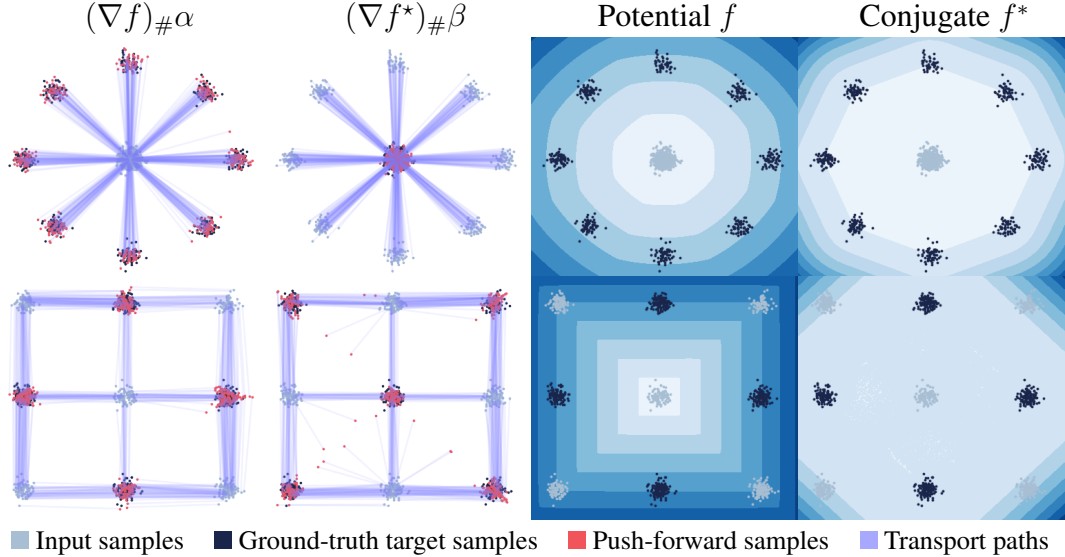

| $(\nabla f)_{\#}\alpha$ | $(\nabla f^\star)_{\#}\beta$ | Potential $f$ | Conjugate $f^*$ |

■ Input samples ■ Ground-truth target samples ■ Push-forward samples ■ Transport paths

Figure 4: Learned potentials on settings considered in Makkuva et al. (2020).

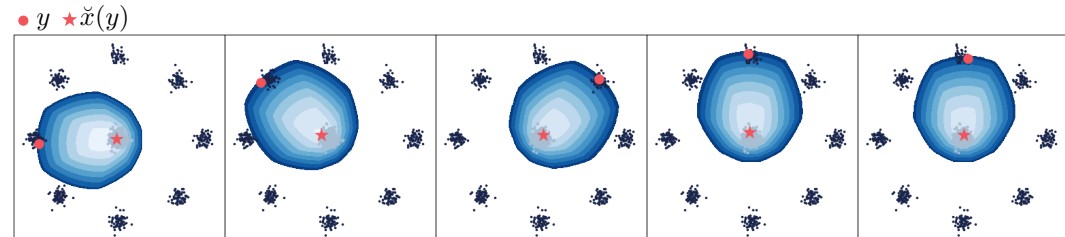

$$\mathcal{G} \qquad \leftarrow ((1-t)I + t\nabla f^\star)_{\#}\mathcal{G} \rightarrow \qquad (\nabla f^\star)_{\#}\mathcal{G}$$

Figure 5: Mesh grid $\mathcal{G}$ warped by the conjugate potential flow $\nabla f^\star$ from the top setting of fig. 4.

• $y$  ★ $\breve{x}(y)$

Figure 6: Sample conjugation landscapes $J(x; y)$ of the top setting of fig. 4. The inverse transport map $\nabla_y f^\star(y) = \breve{x}(y)$ is obtained by minimizing $J$, which is a convex optimization problem. The contour shows $J(x; y)$ filtered to not display a color for values above $J(y; y)$.

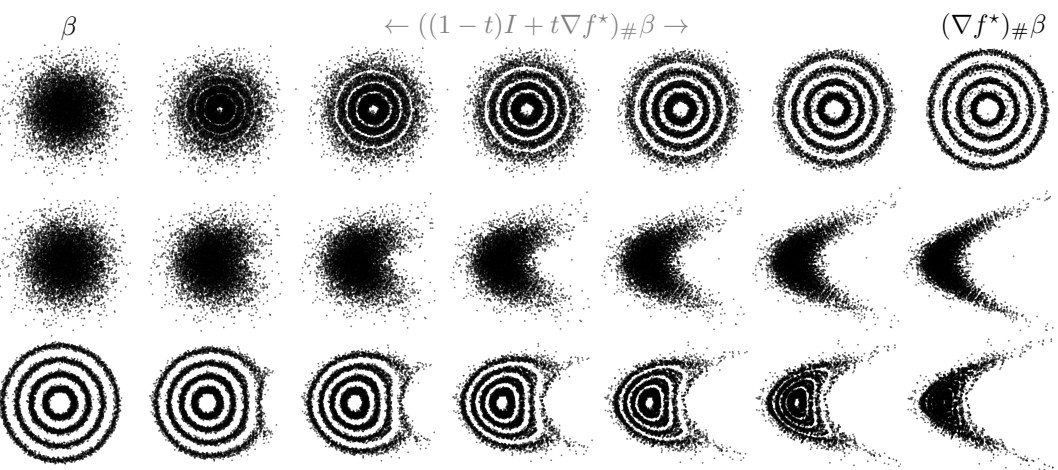

$$\beta \qquad \leftarrow ((1-t)I + t\nabla f^\star)_{\#}\beta \rightarrow \qquad (\nabla f^\star)_{\#}\beta$$

Figure 7: Single-block potential flows on synthetic settings considered in Huang et al. (2020).

ACKNOWLEDGMENTS

I would like to thank Max Balandat, Ricky Chen, Samuel Cohen, Marco Cuturi, Carles Domingo-Enrich, Yaron Lipman, Max Nickel, Misha Khodak, Aram-Alexandre Pooladian, Mike Rabbat, Adriana Romero Soriano, Mark Tygert, and Lin Xiao, for insightful comments and discussions. The core set of tools in Python (Van Rossum and Drake Jr, 1995; Oliphant, 2007) enabled this work, including Hydra (Yadan, 2019), JAX (Bradbury et al., 2018), Flax (Heek et al., 2020), Matplotlib (Hunter, 2007), numpy (Oliphant, 2006; Van Der Walt et al., 2011), and pandas (McKinney, 2012).

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

---

**Algorithm 3** The Broyden-Fletcher-Goldfarb-Shanno (BFGS) method to solve eq. (11) as presented in Nocedal and Wright (1999, alg. 6.1).

---

**Inputs:** Function $J$ to optimize, initial iterate $x_0$ and Hessian approximation $B_0$
$k \leftarrow 0$
**while** unconverged **do**
    Compute the *search direction* $p_k = -B_k^{-1} \nabla_x J_k(x_k)$
    Set $x_{k+1} = x_k + \alpha_k p_k$ where $\alpha_k$ is computed from a *line search* from app. A.2
    Compute $B_k$ with the update in eq. (12)
    $k \leftarrow k + 1$
**end while**
**return** optimal solution $x_k \approx \breve{x}$

---

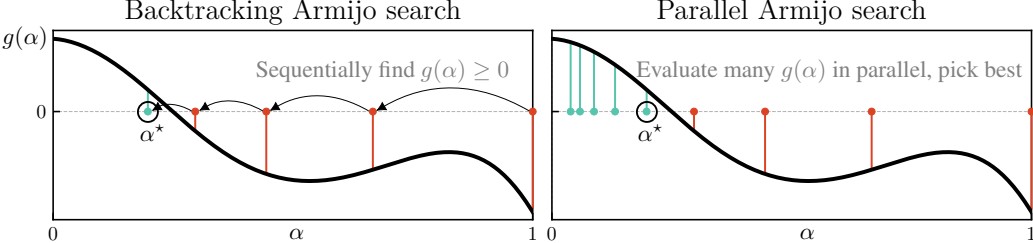

Figure 8: Visualization of backtracking and parallel line searches to solve eq. (17).

## A  L-BFGS OVERVIEW AND LINE SEARCH DETAILS

The conjugate optimization problem in eq. (3) is an unconstrained convex optimization problem for convex potentials, which is a setting BFGS (Broyden, 1970; Fletcher, 1970; Goldfarb, 1970; Shanno, 1970) and L-BFGS (Liu and Nocedal, 1989) methods thrive in. **The default strong Wolfe line search methods in the Jax and JaxOpt L-BFGS implementations may take a long time to solve a batch of optimization problems.** Without efficiently setting the line search method, some of the Wasserstein-2 benchmark experiments in app. C that ran in a few hours would have otherwise taken a *month* to run. This section provides a brief overview of BFGS methods and shows that an Armijo line search can be the most efficient at computing the conjugate.

### A.1  BACKGROUND ON BFGS METHODS

Nocedal and Wright (1999, alg. 6.1) is a standard reference for BFGS methods and extensions and the key steps are summarized in algorithm 3 for solving an optimization problem of the form

$$\breve{x} \in \arg\min_x J(x) \tag{11}$$

where $J : \mathbb{R}^n \to \mathbb{R}$ is possibly non-convex and twice continuously differentiable. The method iteratively finds a solution $\breve{x}$ by 1) maintaining an approximate Hessian around the current iterate, i.e. $B_k \approx \nabla^2 J(x_k)$, 2) computing an approximate *Newton step* $p_k = -B_k^{-1} \nabla_x J_k(x_k)$ using the approximate Hessian, 3) updating the iterate with $x_{k+1} = x_k + \alpha_k p_k$, where $\alpha_k$ is found with a *line search*, and 4) updating the Hessian approximation with the *Sherman–Morrison–Woodbury* formula (Woodbury, 1950; Sherman and Morrison, 1950)

$$B_{k+1} = B_k - \frac{B_k s_s s_k^\top B_k}{s_k^\top B_k s_k} + \frac{y_k y_k^\top}{y_k^\top s_k}, \tag{12}$$

where $y_k = \nabla_x J(x_{k+1}) - \nabla_x J(x_k)$ and $s_k = x_{k+1} - x_k$. Instead of estimating $B_k$ and inverting it in every iteration, most implementations maintain a direct approximation to the *inverse* Hessian $H_k := B_k^{-1}$. The *limited-memory* version of BFGS (L-BFGS) in Liu and Nocedal (1989) propose to replace the inverse Hessian approximation as a *matrix* $H_k$ with the sequence of vectors $[y_k, s_k]$ defining the updates to $H_k$ and never requires instantiating the full $n \times n$ approximation.

---

**Algorithm 4** Backtracking Armijo line search to solve eq. (17)

---

**Inputs:** Iterate $x_k$, search direction $p_k$, decay $\tau$, control parameter $c_1$, initial $\alpha_0$
$\alpha \leftarrow \alpha_{\text{init}}$
**while** $J(x_k + \alpha_j p_k) > J(x_k) + c_1 \alpha_j p_k^\top \nabla_x J(x_k)$ **do**
    $\alpha \leftarrow \tau\alpha$
**end while**
**return** $\alpha$ satisfying the Armijo condition in eq. (17).

---

**Algorithm 5** Parallel Armijo line search to solve eq. (17)

---

**Inputs:** Iterate $x_k$, search direction $p_k$, decay $\tau$, control parameter $c_1$, initial $\alpha_0$, #evaluations $M$
Compute *candidate step lengths* $\alpha_m = \tau^{-m}$ for $m \in [M]$
Evaluate the line search condition $g(\alpha_m)$ from eq. (16) in parallel
**if** all $g(\alpha_m) < 0$ **then**
    `Error: No acceptable step found`
**else**
    **return** largest $\alpha$ satisfying eq. (17), i.e. $\max \alpha_m$ subject to $g(\alpha_m) > 0$
**end if**

---

## A.2 LINE SEARCHES

The *line search* to find the step size $\alpha_k$ for the iterate update $x_{k+1} = x_k + \alpha_k p_k$ is often done with:

1. a *Wolfe* line search (Wolfe, 1969) to satisfy the conditions:

$$J(x_k + \alpha_k p_k) \leq J(x_k) + c_1 \alpha_k p_k^\top \nabla_x J(x_k)$$
$$-p_x^\top \nabla_x J(x_k + \alpha_x p_x) \leq -c_2 p_x^\top \nabla_x J(x_k) \tag{13}$$

where $0 < c_1 < c_2 < 1$,

2. a *strong Wolfe* line search to satisfy the conditions:

$$J(x_k + \alpha_k p_k) \leq J(x_k) + c_1 \alpha_k p_k^\top \nabla_x J(x_k)$$
$$|p_x^\top \nabla_x J(x_k + \alpha_x p_x)| \leq c_2 |p_x^\top \nabla_x J(x_k)| \tag{14}$$

This is often found via the *zoom* procedure from Nocedal and Wright (1999, algorithm 3.5).

3. an *Armijo* line search (Armijo, 1966) to satisfy the first condition:

$$J(x_k + \alpha_k p_k) \leq J(x_k) + c_1 \alpha_k p_k^\top \nabla_x J(x_k). \tag{15}$$

For notational simplicity, we can also write the Armijo condition as:

$$g(\alpha) := f(x_k) + c_1 \alpha p_k^\top \nabla f(x_k) - f(x_k + \alpha p_k) \geq 0 \tag{16}$$

**Remark 14** *The strong Wolfe line search is the most commonly used line search for L-BFGS as it guarantees that the resulting update to the Hessian in eq. (12) stays positive definite, but the Armijo line search may be more efficient as it does not involve re-evaluating the derivative of the objective. Unfortunately, an iterate obtained by an Armijo line search may not satisfy the curvature condition $y_k^\top s_k > 0$ that ensure the Hessian update stays positive definite while a step satisfying the strong Wolfe conditions provably does (Nocedal and Wright, 1999, page 143). Nonetheless, Armijo searches are still combined with BFGS and can be guarded by only updating the Hessian approximation if $y_k^\top s_k > 0$, or a modification thereof, as described in Li and Fukushima (2001); Wan et al. (2012); Fridovich-Keil and Recht (2020) and Berahas et al. (2016, sect. 3.2).*

Table 3: Runtime and number of L-BFGS iterations for line search methods to converge to a solution $\check{x}$ of eq. (11) for conjugating the trained ICNN potential on the 256-dimensional HD benchmark from sect. 5 with a batch of 1024 samples and a tolerance of $\|\nabla J(\check{x})\|_\infty \leq 0.1$, starting from the amortized prediction. The Wolfe and Armijo line search methods use standard values of $c_1 = 10^{-4}$ and $c_2 = 0.9$, all backtracking options use a decay factor of $\tau = 2/3$ with $M = 15$ evaluations, and the runtimes are averaged over 10 trials on an NVIDIA Tesla V100 GPU. In this setting, Armijo line searches without many conditionals or gradient evaluations consistently take the shortest time.

| Base L-BFGS | Line search | Runtime (ms) | # Iterations |
|---|---|---|---|
| Jax | Strong Wolfe Zoom (default) | 4803.40 | 6.21 |
| | Backtracking Armijo | 156.69 | 8.56 |
| | Parallel Armijo | 119.41 | 8.56 |
| JaxOpt | Strong Wolfe Zoom (default) | 776.52 | 7.78 |
| | Backtracking Strong Wolfe | 233.39 | 7.97 |
| | Backtracking Wolfe | 225.30 | 8.65 |
| | Backtracking Armijo | 154.90 | 8.58 |

### A.2.1 THE ARMIJO LINE SEARCH

The *Armijo line search* can be written as the optimization problem

$$\alpha_k(x_k, p_k) = \max \alpha \quad \text{subject to} \quad J(x_k + \alpha p_k) \leq J(x_k) + c_1 \alpha p_k^\top \nabla_x J(x_k). \qquad (17)$$

Eq. (17) is typically solved as shown in algorithm 4 and fig. 8 by setting a *decay factor* $\tau$ and iteratively decreasing a *candidate step length* $\alpha$ until the condition is satisfied.

When the objective $J$ can be efficiently evaluated in parallel on a GPU, and when solving many batches of optimization problems concurrently, e.g. with vmap, the backtracking Armijo line search described in algorithm 4, and the Wolfe line search described in Nocedal and Wright (1999, alg. 7.5), are computationally slowed down by serial and conditional operations. These issues arise from: 1) the sequential nature of the line search, and 2) the fact that the line search may run for a different number of iterations for every optimization problem in the batch. Wolfe line searches such as Nocedal and Wright (1999, alg. 7.5) have other conditionals scoping the search interval that cause the line search to perform potentially different operations for every optimization problem in the batch.

I propose a *parallel* Armijo line search in algorithm 5, which is also visualized in fig. 8, to remove serial and conditional operations to improve the computation of the line search on the GPU for solving batches of optimization problems. The key idea is to instantiate many possible step sizes, evaluate them all at once, and then select the largest $\alpha_m$ satisfying the Armijo condition $g(\alpha_m) \geq 0$.

**Remark 15** *The parallel line search may unnecessarily evaluate more candidate steps sizes than the sequential line search, but on GPU architectures this may not be very detrimental to the performance because additional parallel function evaluations are computationally cheap. Furthermore, when solving a batch of $N$ optimization problems with $M$ line search evaluations, i.e. when using* vmap *on the line search or optimizer, the parallel line search in algorithm 5 can efficiently evaluate $NM$ candidate step lengths in tandem on a GPU and then select the best for each element in the batch.*

**Remark 16** *A potential concern with this parallel line search is that it may not find a step size satisfying the Armijo condition if $M$ is not set to be low enough. While this may be a significant issue for when high-precision solves are needed, I have found in practice for the Euclidean Wasserstein-2 conjugates that taking $M = 10$ line search evaluations frequently finds a solution.*

### A.2.2 COMPARING LINE SEARCH METHODS

Table 3 takes a trained ICNN potential and isolates the comparison between L-BFGS runtimes to only changing the linesearch methods. This is the same optimization procedure and batch size used for all of the training runs on the Wasserstein-2 benchmark. Despite the concerns in remark 14 about the Armijo line search resulting in slower convergence and an indefinite Hessian approximations, the Armijo line searches are consistently able to solve the batch of optimization problems the fastest.

## B  MODEL DEFINITIONS AND PRETRAINING

All of the potential and conjugate amortization models in this paper can be implemented in $\approx 30$–$50$ lines of readable Jax code with Flax (Heek et al., 2020). They are included in the `w2ot/models` directory of the code here, and reproduced here to precisely define them.

### B.1  PRETRAINING AND INITIALIZATION

Following Korotin et al. (2021a), every experimental setting has a pre-training phase so that the potentials and amortization maps approximate the identity mapping, i.e. $\nabla_x f_\theta(x) \approx x$ and $\tilde{x}_\varphi(y) \approx y$.

### B.2  `InitNN`: NON-CONVEX NEURAL NETWORK AMORTIZATION MODEL $\tilde{x}_\varphi$

**Remark 17** *The passthrough on line 18 is helpful for learning an identity initialization.*

```python
class InitNN(nn.Module):
    dim_hidden: Sequence[int]
    act: str = 'elu'

    @nn.compact
    def __call__(self, x):
        assert x.ndim == 2
        n_input = x.shape[-1]

        act_fn = layers.get_act(self.act)

        z = x
        for n_hidden in self.dim_hidden:
            Wx = nn.Dense(n_hidden, use_bias=True)
            z = act_fn(Wx(z))

        Wx = nn.Dense(n_input, use_bias=True)
        z = x + Wx(z) # Encourage identity initialization.

        return z
```

## B.3 ICNN: INPUT-CONVEX NEURAL NETWORK POTENTIAL $f_\theta$

`actnorm` is the *activation normalization* layer from Kingma and Dhariwal (2018), which was also used in the ICNN potentials in Huang et al. (2020) and normalizes the activations at initialization to follow a normal distribution.

**Remark 18** *Applying an activation to the output on line 41 is helpful to lower-bound the otherwise unconstrained potential and adds stability to the training.*

**Remark 19** *The final quadratic on line 46 makes it easy to initialize the potential to the identity.*

**Remark 20** *This ICNN does **not** use the quadratic activations proposed in Korotin et al. (2019, Appendix B.1). While I did not heavily experiment with them, table 1 shows that this ICNN architecture without the quadratic activations performs better than the results originally reported in Korotin et al. (2021a) which use an ICNN architecture with the quadratic activations.*

```python
class ICNN(nn.Module):
    dim_hidden: Sequence[int]
    act: str = 'elu'
    actnorm: bool = True

    def setup(self):
        kernel_init = nn.initializers.variance_scaling(
            1., "fan_in", "truncated_normal")
        num_hidden = len(self.dim_hidden)

        w_zs = list()
        for i in range(1, num_hidden):
            w_zs.append(layers.PositiveDense(
                self.dim_hidden[i], kernel_init=kernel_init))
        w_zs.append(layers.PositiveDense(1, kernel_init=kernel_init))
        self.w_zs = w_zs

        w_xs = list()
        for i in range(num_hidden):
            w_xs.append(nn.Dense(
                self.dim_hidden[i], use_bias=True,
                kernel_init=kernel_init))

        w_xs.append(nn.Dense(1, use_bias=True, kernel_init=kernel_init))
        self.w_xs = w_xs

    @nn.compact
    def __call__(self, x):
        assert x.ndim == 2
        n_input = x.shape[-1]
        act_fn = layers.get_act(self.act)

        z = act_fn(self.w_xs[0](x))
        for Wz, Wx in zip(self.w_zs[:-1], self.w_xs[1:-1]):
            z = Wz(z) + Wx(x)
            if self.actnorm:
                z = layers.ActNorm()(z)
            z = act_fn(z)

        y = act_fn(self.w_zs[-1](z) + self.w_xs[-1](x))
        y = jnp.squeeze(y, -1)

        log_alpha = self.param(
            'log_alpha', nn.initializers.constant(0), [])
        y += jnp.exp(log_alpha)*0.5*utils.batch_dot(x, x)

        return y
```

### B.4 `PotentialNN`: NON-CONVEX NEURAL NETWORK (MLP) POTENTIAL $f_\theta$

**Remark 21** *Consistent with remarks 18 and 19, applying an activation to the last layer (line 18) and adding a final quadratic term (line 24) helps this non-convex potential model too.*

```python
class PotentialNN(nn.Module):
    dim_hidden: Sequence[int]
    act: str = 'elu'

    @nn.compact
    def __call__(self, x):
        assert x.ndim == 2
        n_input = x.shape[-1]

        act_fn = layers.get_act(self.act)

        z = x
        for n_hidden in self.dim_hidden:
            Wx = nn.Dense(n_hidden, use_bias=True)
            z = act_fn(Wx(z))

        Wx = nn.Dense(1, use_bias=True)
        z = act_fn(Wx(z))

        z = jnp.squeeze(z, -1)

        log_alpha = self.param(
            'log_alpha', nn.initializers.constant(0), [])
        z += 0.5*jnp.exp(log_alpha)*utils.batch_dot(x, x)

        return z
```

### B.5 ConvPotential: Non-convex convolutional potential $f_\theta$

**Remark 22** *I was not able to easily add batch normalization (Ioffe and Szegedy, 2015) to this potential. In contrast to standard use cases of batch normalization that only call into a batch-normalized model once over samples from a single distribution, the dual objective in eq. (4) calls into the potential multiple times to estimate $\mathbb{E}_{x\sim\alpha}\, f_\theta(x)$ and $\mathbb{E}_{y\sim\beta}\, f_\theta(\breve{x}(y))$, which also involve internally solving the conjugate optimization problem in eq. (3) to obtain $\breve{x}$. This makes it not clear what training and evaluation statistics batch normalization should use when computing the dual objective. One choice could be to only use the statistics induced from the samples $x\sim\alpha$.*

```python
class ConvPotential(nn.Module):
    act: str = 'elu'

    mean = jnp.array([0.485, 0.456, 0.406])
    std = jnp.array([0.229, 0.224, 0.225])

    @nn.compact
    def __call__(self, x):
        assert x.ndim == 2 # Images should be flattened
        num_batch = x.shape[0]

        x_flat = x # Save for taking the quadratic at the end.

        # Reshape and renormalize
        x = x.reshape(-1, 3, 64, 64).transpose(0, 2, 3, 1)
        x = (x + 1.)/2.
        x = (x-self.mean) / self.std
        y = x

        act_fn = layers.get_act(self.act)

        conv = nn.Conv(128, kernel_size=[4,4], strides=2)
        y = act_fn(conv(y))

        conv = nn.Conv(128, kernel_size=[4,4], strides=2)
        y = act_fn(conv(y))

        conv = nn.Conv(256, kernel_size=[4,4], strides=2)
        y = act_fn(conv(y))

        conv = nn.Conv(512, kernel_size=[4,4], strides=2)
        y = act_fn(conv(y))

        conv = nn.Conv(1024, kernel_size=[4,4], strides=2)
        y = act_fn(conv(y))

        conv = nn.Conv(
            1, kernel_size=[2,2], padding='VALID', strides=1)
        y = act_fn(conv(y))
        y = y.squeeze([1,2,3])

        assert y.shape == (num_batch,)

        log_alpha = self.param(
            'log_alpha', nn.initializers.constant(0), [])
        y += 0.5*jnp.exp(log_alpha)*utils.batch_dot(x_flat, x_flat)

        return y
```

# C    ADDITIONAL WASSERSTEIN-2 BENCHMARK EXPERIMENT DETAILS

## C.1    HYPER-PARAMETERS

Tables 4 and 5 detail the main hyper-parameters for the Wasserstein-2 benchmark experiments. I tried to keep these consistent with the choices from Korotin et al. (2021a), e.g. using the same batch sizes, number of training iterations, and hidden layer sizes for the potential.

All experiments use the same settings for the conjugate solvers: The conjugate solvers stop early if all dimensions of the iterates change by less than 0.1, and otherwise run for a maximum of 100 iterations. The line search parameters for the parallel Armijo search in algorithm 5 for L-BFGS are to decay the steps with a base of $\tau = 1.5$ and to search $M = 10$ step sizes. With the Adam conjugate solver, I use the default $\beta = [0.9, 0.999]$ with an initial learning rate of 0.1 with a cosine annealing schedule to decrease it to $10^{-5}$.

Table 4: Hyper-parameters for the $D$-dimensional Wasserstein-2 benchmark experiments

| Name | Value |
|---:|:---|
| potential model $f_\theta$ | `ICNN` or `PotentialNN` |
| $f_\theta$ hidden layer sizes | $[\max(2D, 64), \max(2D, 64), \max(D, 32)]$ |
| conjugate amortization model $\tilde{x}_\varphi$ | `InitNN(dim_hidden=[512, 512])` |
| activation functions | ELU (Clevert et al., 2015) |
| # training iterations | 250000 |
| optimizer | Adam with cosine annealing ($\alpha$=1e-4) |
| initial learning rate | 5e-4 |
| Adam $\beta$ | [0.5, 0.5] |
| batch size | 1024 |

Table 5: Hyper-parameters for the CelebA64 Wasserstein-2 benchmark experiments

| Name | Value |
|---:|:---|
| potential model $f_\theta$ | `ConvPotential` |
| conjugate amortization model $\tilde{x}_\varphi = \nabla g_\varphi$ | Gradient of `ConvPotential` |
| activation functions | ELU (Clevert et al., 2015) |
| number of training iterations | 50000 |
| optimizer | Adam with cosine annealing ($\alpha$=1e-4) |
| initial learning rate | 1e-3 |
| Adam $\beta$ | [0.5, 0.5] |
| batch size | 64 |

## C.2 Convergence of L-BFGS and Adam for solving the conjugate

Figure 9 shows that with a non-convex potential, many of the initial amortized predictions are suboptimal and difficult for the L-BFGS to improve upon. This indicates that the amortized predictions may be in parts of the space that are difficult to recover from and suggests a future avenue of work better characterizing and recovering from this behavior. L-BFGS converges fast to an optimal solution in fig. 10 while Adam often gets stuck at suboptimal solutions.

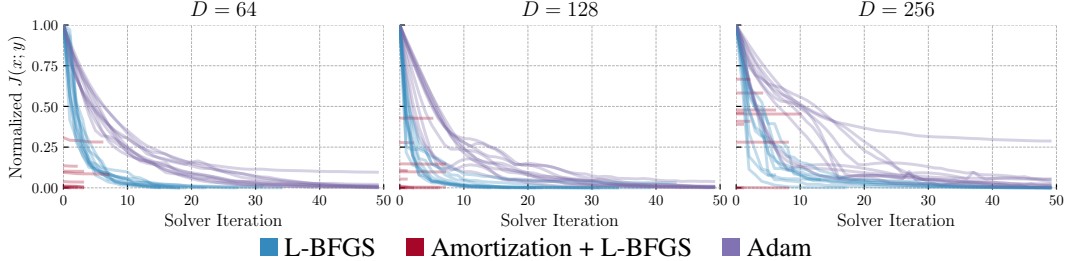

Figure 9: Conjugate solver convergence on the HD benchmarks with a NN potential.

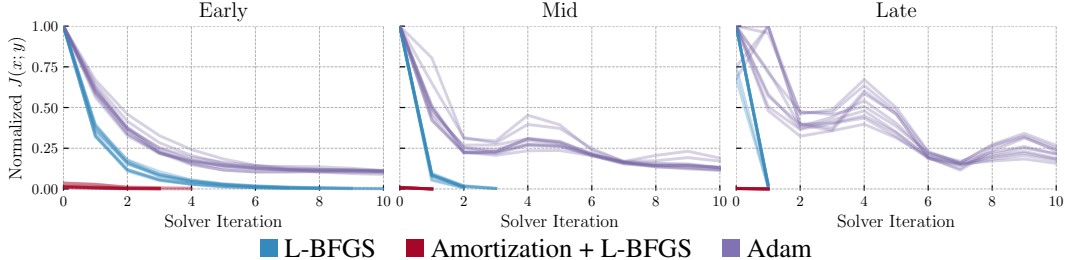

Figure 10: Conjugate solver convergence on the CelebA64 benchmark

## C.3 Additional runtimes and conjugation information

Tables 6 and 7 contain additional experimental information with the:

1. **wall-clock time** for the entire training run measured on an NVIDIA Tesla V100 GPU,

2. **number of conjugation iterations** from algorithm 2 for the conjugate solver to converge at the end of training after warm-starting it from the conjugate amortization model's prediction, and

3. **runtime for the conjugate solver** to converge on a batch of instances, set to the batch size used during training, i.e. 1024 for the HD benchmark and 64 for the CelebA64 benchmark.

These results give an idea of how much additional time is spent fine-tuning. On the HD benchmark, fine-tuning takes between $\approx$ 10–50ms per batch. The overall wall clock time may take $\approx$ 2–3 times longer than the training runs without fine-tuning, but are able to find significantly better solutions. On the CelebA64 benchmarks, the conjugation time impacts the overall runtime even less because, especially in the "Mid" and "Late" settings as the transport maps here are close to being the identity mapping and are easy to conjugate.

**Remark 23** *Some settings immediately diverged to an irrecoverable state providing a $\mathcal{L}^2$-UVP of $10^9$, including runs using the objective-based and cycle amortization losses without fine-tuning. I early-stopped those experiments and do not report the runtimes or conjugation times here, as the few minutes that the objective-based amortization experiments took to diverge is not very interesting or comparable to the times of the experiments that converged.*

Table 6: Additional runtime and conjugation information for the HD benchmark. These report the median time from the converged runs.

| | Amortization loss | Conjugate solver | Runtime (hours) | | | Final conjugation iter | | | Conj runtime (seconds) | | |
|---|---|---|---|---|---|---|---|---|---|---|---|
| | | | $n = 64$ | $n = 128$ | $n = 256$ | $n = 64$ | $n = 128$ | $n = 256$ | $n = 64$ | $n = 128$ | $n = 256$ |
| ICNN | Cycle | None | 1.10 | 1.24 | 1.58 | | | | | | |
| ICNN | Cycle | L-BFGS | 4.39 | 7.76 | 21.14 | 14.32 | 19.38 | 70.23 | 0.05 | 0.10 | 0.29 |
| ICNN | Objective | L-BFGS | 2.69 | 4.70 | 13.86 | 5.43 | 6.96 | 8.13 | 0.02 | 0.05 | 0.13 |
| ICNN | Regression | L-BFGS | 2.74 | 4.30 | 12.53 | 5.64 | 6.34 | 8.73 | 0.02 | 0.04 | 0.11 |
| ICNN | Cycle | Adam | 2.09 | 2.80 | 0.86 | 37.23 | 51.34 | 97.21 | 0.02 | 0.03 | 0.06 |
| ICNN | Objective | Adam | 2.23 | 2.99 | 4.94 | 29.44 | 38.81 | 55.05 | 0.02 | 0.03 | 0.05 |
| ICNN | Regression | Adam | 2.09 | 2.86 | 4.81 | 29.51 | 38.20 | 54.21 | 0.02 | 0.03 | 0.06 |
| NN | Objective | L-BFGS | 2.31 | 3.34 | 6.05 | 5.05 | 6.50 | 5.77 | 0.02 | 0.03 | 0.05 |
| NN | Regression | L-BFGS | 2.28 | 3.21 | 5.76 | 5.21 | 4.72 | 5.01 | 0.02 | 0.02 | 0.05 |
| NN | Objective | Adam | 1.61 | 2.18 | 3.70 | 27.84 | 36.81 | 45.82 | 0.01 | 0.02 | 0.04 |
| NN | Regression | Adam | 1.77 | 2.51 | 3.79 | 28.28 | 31.46 | 40.31 | 0.01 | 0.02 | 0.04 |

Table 7: Additional runtime and conjugation information for the CelebA64 benchmark

| Amortization loss | Conjugate solver | Model | Direction | Runtime (hours) | | | Final conj iter | | | Conj runtime (seconds) | | |
|---|---|---|---|---|---|---|---|---|---|---|---|---|
| | | | | Early | Mid | Late | Early | Mid | Late | Early | Mid | Late |
| Objective | None | Conv | Forward | 3.28 | 3.41 | 3.28 | | | | | | |
| Cycle | None | Conv | Forward | 0.94 | 4.17 | 4.17 | | | | | | |
| Cycle | Adam | Conv | Forward | 5.44 | 4.78 | 3.57 | 18.23 | 2.02 | 2.80 | 0.15 | 0.11 | 0.04 |
| Cycle | L-BFGS | Conv | Forward | 6.38 | 3.79 | 3.74 | 5.18 | 2.00 | 2.00 | 0.22 | 0.04 | 0.04 |
| Objective | Adam | Conv | Forward | 5.40 | 4.85 | 3.47 | 19.87 | 1.83 | 1.79 | 0.17 | 0.13 | 0.03 |
| Objective | L-BFGS | Conv | Forward | 6.09 | 3.86 | 3.70 | 4.48 | 2.01 | 2.00 | 0.21 | 0.06 | 0.04 |
| Regression | Adam | Conv | Forward | 5.44 | 4.86 | 3.51 | 22.12 | 2.84 | 1.01 | 0.19 | 0.14 | 0.02 |
| Regression | L-BFGS | Conv | Forward | 5.96 | 3.77 | 3.66 | 4.55 | 2.01 | 2.02 | 0.22 | 0.04 | 0.04 |

## D  ADDITIONAL 2D DEMONSTRATION EXPERIMENT DETAILS

Table 8 details the main hyper-parameters for the synthetic benchmark experiments, and fig. 11 shows additional conjugation landscapes.

**Remark 24** *I found leaky ReLU activations on the potential model to work better in these low-dimensional settings than ELU activations, which work better in the HD benchmark settings. I do not have a strong explanation for this but found the LReLU capable of performing sharper transitions in the space, e.g. the sharp boundaries shown in fig. 4. One reason that the ELU potentials could perform better on the benchmark settings is that the ground-truth transport maps in the benchmark, described in Korotin et al. (2021a, Appendix B.1), use an ICNN with CELU activations (Barron, 2017) which may be easier to recover with potential models that use ELU activations.*

I trained convex and non-convex potentials on every synthetic setting and show the results from the best-performing potential model, which are:

- Makkuva et al. (2020): an ICNN. This setting originally considered convex potentials, and the non-convex potentials I tried training on these settings diverged,

- Rout et al. (2021): a non-convex potential (an MLP). This setting also originally considered an MLP and I couldn't find an ICNN that accurately transports between the highly curved and concentrated parts of the measures.

- Huang et al. (2020): a non-convex potential (an MLP). **In contrast to the ICNNs origi-nally used, I found that an MLP works better when learned with the OT dual.** Almost every setting in Huang et al. (2020) requires composing multiple blocks of ICNNs, which means the flow will not necessary be the optimal transport flow, while the non-convex MLP potential I am using here estimates the *optimal* transport map between the measures.

All of the synthetic settings use the L-BFGS conjugate solver set to obtain slightly higher precision solves than in the Wasserstein-2 benchmark. The conjugate solver stops early if all dimensions of the iterates change by less than 0.001, and otherwise run for a maximum of 100 iterations. The line search parameters for the parallel Armijo search in algorithm 5 for L-BFGS are to decay the steps with a base of $\tau = 1.5$ and to search $M = 30$ step sizes.

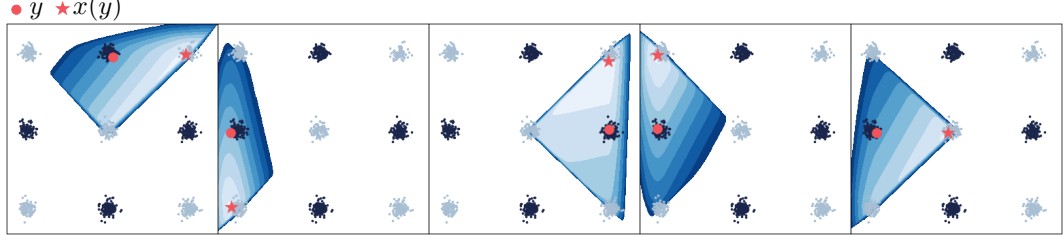

Figure 11: Sample conjugation landscapes $J(x; y)$ of the bottom setting of fig. 4. The inverse transport map $\nabla_y f^\star(y) = \breve{x}(y)$ is obtained by minimizing $J$, which is a convex optimization problem. The contour shows $J(x; y)$ filtered to not display a color for values above $J(y; y)$.

Table 8: Hyper-parameters for the synthetic experiments

| Name | Value |
|---|---|
| potential model $f_\theta$ | `ICNN` or `PotentialNN` |
| $f_\theta$ hidden layer sizes | [128, 128] |
| conjugate amortization model $\tilde{x}_\varphi$ | `InitNN(dim_hidden=[512, 512])` |
| activation functions | Leaky ReLU with slope 0.2 |
| # training iterations | 50000 |
| optimizer | Adam with cosine annealing ($\alpha$=1e-4) |
| initial learning rate | 5e-4 |
| Adam $\beta$ | [0.5, 0.5] |
| batch size | 10000 |

### D.1 Non-convex regions in the learned potentials

*Brenier's theorem* (Brenier, 1991) shows that the known Wasserstein-2 optimal transport map associated with the negative inner product cost is the gradient of a convex function, i.e. $\breve{T}(x) = \nabla_x \hat{f}(x)$. Because of this, optimizing over convex potentials is theoretically nice and also results in a convex and easy conjugate optimization problem in eq. (3) to compute $f^\star$. The input-convex property is usually enforced by constraining all of the weights of the network to be positive in every layer except the first. Unfortunately, in practice, the positivity constraints of a convex potential may be prohibitive and not easy to optimize over and result in sub-optimal transport maps. In other words, the parameter optimization problem over the input-convex model is still non-convex and may be exasperated by the input-convex constraints. Due to these limitations, non-convex potentials are appealing as their parameter space is less constrained and may therefore be easier to search over. And in practice, this has been shown to be true, e.g. the main results in table 1 show that a non-convex potential significantly outperforms the convex potential. However, non-convex potentials can result in non-convex conjugate optimization problems in eq. (3) that can cause significant numerical instabilities and an exploding upper-bound on the dual objective.

Figure 12 illustrates a small non-convex region arising in a learned non-convex potential. While the non-convex region mostly does not impact the transport map in this case, they can easily blow up and make the dual optimization problem challenging. In contrast, the ICNN-based convex potential provably retains convexity and keeps this region nicely flat, but the constraints on the parameter space may hinder the performance.

Interpolation from a non-convex potential (an MLP)

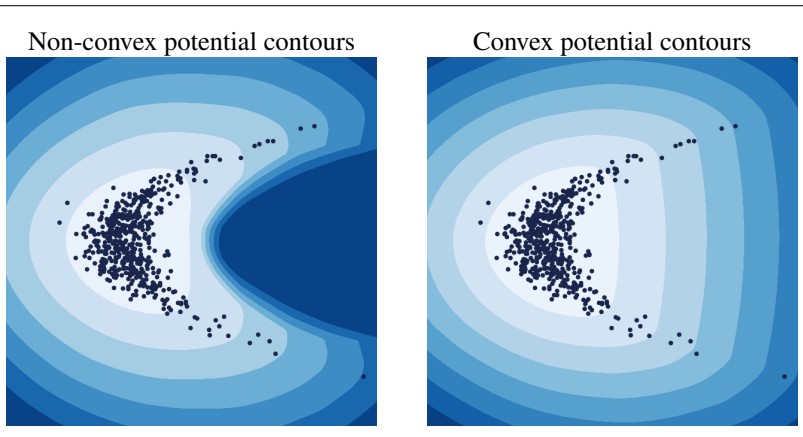

Figure 12: Convex and non-convex potentials trained on the same transport task.

## D.2    INTERPOLATIONS ON SYNTHETIC SETTINGS FROM ROUT ET AL. (2021)

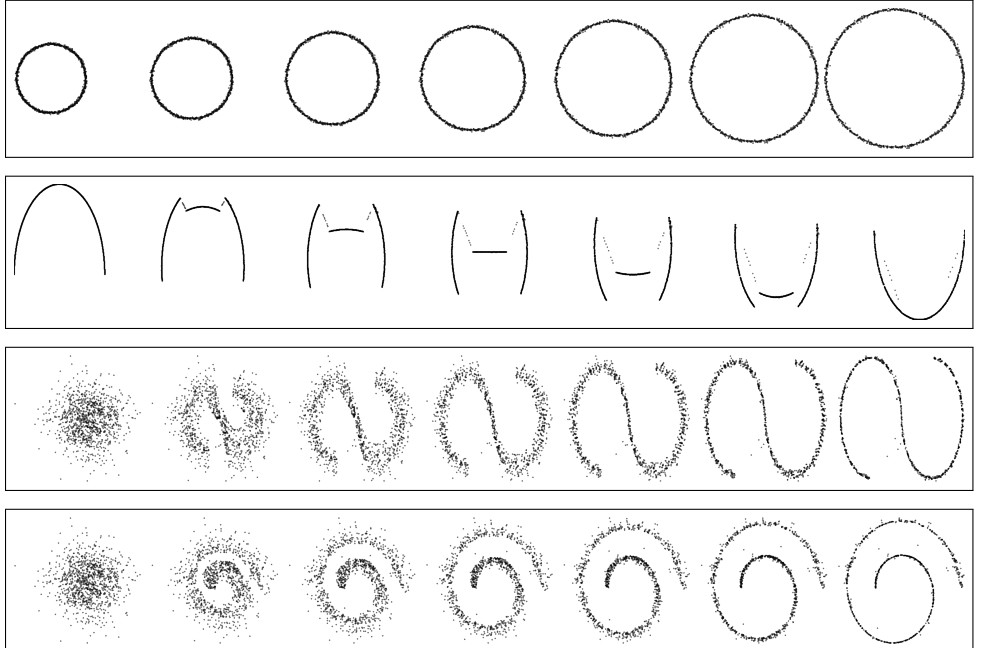

