# OpenReview forum: "On amortizing convex conjugates for optimal transport"
_ICLR.cc/2023/Conference — ICLR 2023 poster_

### Official Review · Reviewer_G2nL · 2022-10-20

**Confidence:** 5
**Correctness:** 4
**Technical Novelty And Significance:** 2
**Empirical Novelty And Significance:** 4
**Recommendation:** 6

**Clarity, Quality, Novelty And Reproducibility:**

The clarity of the paper is very good and sufficient code snippets and algorithm boxes were provided for reproducibility. I personally like the style of using bold text. For novelty, see the comments above.


**Strength And Weaknesses:**

## Strengths:
* The problem of computing the conjugate map effectively is a central one in neural OT.
* The proposed fine-tuning step is simple in concept/implementation yet very effective.
* Careful software engineering is done to obtain good experimental results.
* Lots of good empirical advice from the remarks in the paper for OT practitioners.


## Weaknesses:
* The novelty of the proposed method is quite limited. There is no new methodological formulation or new theoretical results. To me, it seems mostly like a nice engineering hack (which works quite well).
* The novelty of the new line search method is also limited. The proposed line search improvement is a straightforward way to convert a classical sequential algorithm involving conditional branches to a batched version.
* In a couple of the remarks there are some new perspectives (e.g. Remark 5, 7) of viewing existing formulation in different ways, but I do not find these new perspectives reveal particularly interesting new insights.
* The paper does not reveal a new understanding of why certain combinations of parts of the algorithms work better than others. Some heuristics are provided but could be better explained (e.g. Remark 8 and Figure 1).

## Detailed comments:
* When citing Brenier's theorem below (3), it should be mentioned that it only holds when one distribution has a density (which I believe is the assumption for this paper and most of the related works).
* The message of Remark 8 and Figure 1 is not clear to me. Does it just mean solving (9) is easier than non-convex (6) and (8)?
* In Table 2, ICNN performs very poorly compared to MLP, even with fine-tuning. What could be an explanation for this, other than ICNN is just hard to train as commonly understood? I figure fine-tuning using regression loss should make ICNN easier to train.



**Summary Of The Paper:**

This paper suggests fine-tuning a learnable conjugate map when solving the Wasserstein-2 distance dual formulation (where the potential map is parameterized using a neural network) can lead to significant improvement of the resulting transport map as demonstrated on the Wasserstein-2 benchmark by Korotin et al. (2021a). To facilitate the optimization during fine-tuning, a new parallel line search for L-BFGS is implemented in Jax which is much faster than Jax's default line search method.


**Summary Of The Review:**

Overall I think this is a good paper, despite not having a lot of novelty as mentioned above. The experiments are carefully done. OT practitioners can all learn something useful from reading this paper.

---

> ### Author Response · Authors · 2022-11-18
> **Response part 1/2**
>
> Thanks very much for reviewing my paper! I am happy to hear that you enjoyed reading it and think that OT practitioners can learn something useful from it.  :-)
>
> I’ve posted an updated version of the paper and here are some more responses inline to your comments and questions:
>
> > The novelty of the proposed method is quite limited. There is no new methodological formulation or new theoretical results. To me, it seems mostly like a nice engineering hack (which works quite well). [...] In a couple of the remarks there are some new perspectives (e.g. Remark 5, 7) of viewing existing formulation in different ways, but I do not find these new perspectives reveal particularly interesting new insights.
>
> I agree that the paper is mostly building on the theoretical foundations of other papers, such as Amos (2022), Taghvaei and Jalali (2019), Makkuva et al. (2020), and Korotin et al. (2019). To me, the contribution of my submission here is to connect these all together, which is shown in Remarks 3-8. I can further expand on any of these if it would be helpful, but at least to me, the connections I am making in these remarks are not evident from the original sources. I was especially surprised to realize the results in Section 3.2.2 showing that the cycle consistency term in the W2GN method of Korotin et al. (2019) can be derived from amortizing the first-order optimality conditions of the conjugate. This is novel from the perspective of W2GN, which does not take the amortization perspective, and also from Amos (2022), which does not discuss much the idea of amortizing the optimality conditions.
>
> > The novelty of the new line search method is also limited. The proposed line search improvement is a straightforward way to convert a classical sequential algorithm involving conditional branches to a batched version.
>
> I agree the parallel line search is not very novel. I have moved all of the parallel line search details to the appendix and have toned down the writing everywhere. Please let me know if there’s anything else on the presentation of the line search details that I can improve!
>
> > The paper does not reveal a new understanding of why certain combinations of parts of the algorithms work better than others. Some heuristics are provided but could be better explained (e.g. Remark 8 and Figure 1).
>
> Theoretically understanding amortization choices and approximations to the conjugate has also been elusive in the prior work, e.g., [the ICLR 2021 reviews of Korotin et al. (2019)](https://openreview.net/forum?id=bEoxzW_EXsa&noteId=yRMmEAZB2Q8) also point out that their theory assumes oracle solutions to the optimization problems. I am unfortunately not too sure how to go about formulating the right theoretical statements or proofs necessary to make insightful experimental choices here. I will think a little more about the right way of stating this in the paper and will expand upon the “limitations” section to point the interested readers to potentially interesting open theoretical questions here.
>
> One key takeaway and understanding that I hope my paper does add to the community is the simple fact that a better approximation to the conjugate with fine-tuning significantly helps the performance. All of the other papers in this space, e.g. Nham Dam et al. (2019), Makkuva et al. (2020), Korotin et al. (2019) consider approximating the conjugate but do not mention or experiment with the idea of fine-tuning their approximations. I think fine-tuning in these settings is extremely reasonable and something that should always be considered, and my paper shows that this can result in a ~200-400% improvement of the transport maps on the Wasserstein-2 benchmark presented at NeurIPS 2021.

---

> > ### Author Response · Authors · 2022-11-18
> > **Response part 2/2**
> >
> >
> > > When citing Brenier's theorem below (3), it should be mentioned that it only holds when one distribution has a density (which I believe is the assumption for this paper and most of the related works).
> >
> > Thank you, I have clarified this.
> >
> > > The message of Remark 8 and Figure 1 is not clear to me. Does it just mean solving (9) is easier than non-convex (6) and (8)?
> >
> > In remark 8 and Figure 1, my intended message was to point out that the cycle consistency term may be problematic for non-convex functions because satisfying the first-order optimality conditions is not sufficient for optimality. The prior papers using cycle consistency, e.g. the W2GN one, do not make this connection. I will continue thinking about this fact, as reviewer 7Bdg has also commented that the contents of this remark should be evident.
> >
> > And on the convexity/non-convexity of (6), (8), and (9), I do not think that the convexity with respect to the model’s prediction is the right part to focus on there because they all are still non-convex with respect to the parameters of the amortization model. I think the figure provides some intuition on how the amortization loss choice of Makkuva et al., (2020) compares to the cycle choice of Korotin et al. (2019), and how those compare to just onto the optimal solution. While this mechanically shows an example, I hesitate to use it, or any of the results, to say which of these choices is better and instead have tried to defer any comparisons between them to the experimental results using all of them.
> >
> > > ICNN performs very poorly compared to MLP, even with fine-tuning. What could be an explanation for this, other than ICNN is just hard to train as commonly understood? I figure fine-tuning using regression loss should make ICNN easier to train.
> >
> > I originally started this project only looking at ICNN-based potentials because it seems reasonable that their universal representation capacity would be able to find the optimal potential, which is also what the Convex Potential Flows paper by Huang et al. (2020) argues. However it seems like in practice the constrained parameterization space of an ICNN (non-negative constraints on the weights in every layer except the first) are not as easy to search over or represent functions as the unconstrained parameterization space of an MLP, even when the conjugate showing up in the objective is exactly computed. One last note about MLP potentials is that the dual formulation in equation (2) is only over Lebesgue-integrable functions and while Brenier’s theorem gives that the *solution* is a convex function, it’s okay to search over a larger space of non-convex functions.

---

> > > ### Comment · Reviewer_G2nL · 2022-11-23
> > > **Response to author**
> > >
> > > Thank you for your detailed and thoughtful response. The revision has resolved most of my comments regarding clarity. Some additional comments:
> > > * I like the idea of amortization a lot and empirically it is quite effective. I suspect there are deeper theoretical questions that could help explain why it is so helpful. My intuition is the following. The loss of amortization (9) is very nice and convex (with respect to the output of the neural net). There have been many machine learning theories that show global convergence for such cases (e.g. https://arxiv.org/abs/1811.03962). On the other hand, cycle consistency loss (8) is highly non-convex (there is the composition of the network of $f$ and the network of $\tilde x_\varphi$, and another gradient is taken). Moreover, second-order derivatives are needed when optimizing (8), which can be more challenging if activation functions are not smooth enough (and if they are too smooth they might not be expressive enough). In a sense, amortization helps guide the difficult optimization (8) to reach a better minimum.
> > >
> > > * In general, minimizing an $\ell^2$-loss of the optimality condition can be inefficient compared to smarter alternatives. Another example that came to mind (very unrelated) is the deep Ritz method https://arxiv.org/pdf/1710.00211.pdf versus PINN (https://en.wikipedia.org/wiki/Physics-informed_neural_networks).
> > >
> > > * Regarding ICNN being difficult to optimize, I think an alternative way to parameterize a convex function is the following (I cannot find the source). Consider $g(x) = \frac{\lambda}{2} x^T x - f(x)$ where $f$ has $\lambda$-Lipschitz gradient. This is sufficient for $g$ to be convex. Then to ensure $f$ has $\lambda$-Lipschitz gradient, you can add an additional loss term to make this happen. This could be an alternative to ICNN.

---

> > > > ### Author Response · Authors · 2022-11-23
> > > > **Response**
> > > >
> > > > > the loss of amortization (9) is very nice and convex [... o]n the other hand, cycle consistency loss (8) is highly non-convex
> > > >
> > > > I agree that the regression in eq. (9) may be much nicer than the cycle consistency term eq. (8) for these reasons. It's also interesting to think about the objective loss in eq. (6). It's in general non-convex, but may be desirable as it more directly optimizes for the objective value of the prediction, meaning that an inaccurate prediction may still have a reasonable objective value.  Most of the other successes of amortization (VAEs, RL policy learning) optimize directly for the objective like in eq. (6).
> > > >
> > > > > In general, minimizing an $\ell^2$-loss of the optimality condition can be inefficient compared to smarter alternatives.
> > > >
> > > > I don't immediately understand the difference between deep Ritz and PINNs or how those connect here, but I noticed another interesting parallel when reading the [Physics-Informed Neural Operator for Learning Partial Differential Equations](https://arxiv.org/abs/2111.03794) (PINO) paper. The PDE loss in their eq (3) is optimizing for the $\ell^2$ terms of the PDE residuals/optimality conditions and the data loss in their eq (5) is doing an $\ell^2$ regression onto the known solution.
> > > >
> > > > > Regarding ICNN being difficult to optimize, I think an alternative way to parameterize a convex function
> > > >
> > > > That's interesting! I've not seen that formulation before. Although if it also requires penalizing $f$ to have a $\lambda$-Lipschitz gradient, then it seems worth considering a method that just penalizes convexity directly, unless if the Lipschitz gradient penalty is for some reason better than a convexity penalty. I thought about ablating the potential architectures/penalties more in this paper but thought it may be too distracting from the paper's main focus on amortization. I'm also curious to try the input convex maxout network (ICMN) from [this paper](https://arxiv.org/abs/2205.13684) as it seems like the ICNN parameterization was very limiting for the OT problems there. It would be relatively easy to try these ablations out as new architectures and penalties should be pretty easy to add to the code (which now is also attached to this submission) and all of the experiments can be re-run with a few commands.

---

### Official Review · Reviewer_dkEA · 2022-10-24

**Confidence:** 2
**Correctness:** 3
**Technical Novelty And Significance:** 3
**Empirical Novelty And Significance:** 3
**Recommendation:** 6

**Clarity, Quality, Novelty And Reproducibility:**

The paper is structured nicely. The presentation and writing of the paper could be improved a little bit to facilitate readers not very familiar with the area, and save sometime from memorizing all the equations and settings in different prior works while reading the paper. The paper has done a lot of comparison with prior works all over the paper and maybe doing so more systematically will help the presentation too.

In terms of novelty, the idea is interesting yet the theoretical contribution over prior works is not very clear (see comments above).

**Strength And Weaknesses:**

This paper studies an important problem in OT and provides interesting idea and compelling practical implementation of the proposed method. The experiment part looks solid and convincing. The theoretical part feels less convincing for the following reasons:
- If not mistaken, it feels the key idea of amortizing largely follows the amortizing optimization framework in Amos [2022] and other prior work, and the main part of Section 3 is mostly restating this framework in the particular setting of OT. Most new designs tailored to the setting are discussed in Section 4, but most of it is also still using well known optimizer like Adam and L-BFGS to directly solve the Conjugate function. In that case, the contribution of the theoretical part of the paper reads unclear to me.
- Though I am unsure possible or not, it may be helpful to have at least some convergence guarantee, stability analysis, or even hard instance which explicitly shows why *all methods* need to be sensitive to model's hyper-parameters.
- There are many remarks in the paper, some of them seem a bit vague and in particular, require a very good knowledge on previous work to be able to understand, e.g. Remark 6, Remark 11. It may be helpful to discuss their formulations first and then distinction to this work in detail in the main text.

**Summary Of The Paper:**

The paper studies efficient method for computing convex conjugate arising in wasserstein OT problem. Although exactly and approximately computing so is believed to be hard in prior work, this work proposes a new method based on amortized approximation scheme could be used to computing the exact conjugate. This amortization with fine-tuning also has favorable practical performance, improving over all Wasserstein-2 benchmark tasks, and produce stable results on synthetic dataset.

**Summary Of The Review:**

I am unfamiliar with this area and especially for the computational perspective for 2-wasserstein OT. The paper reads like trying to address an important problem, and the practical performance is quite compelling. My slight concerns are regarding the novelty of the algorithmic idea, lack of support in theory, and a potential for a better presentation. I'd love to hear from the author's perspective before giving a final assessment from my side.

---

> ### Author Response · Authors · 2022-11-18
> **Response**
>
> Thanks very much for the review of the paper!  I’ve posted an updated version of the paper and here are some more responses inline to your comments and questions:
>
> > If not mistaken, it feels the key idea of amortizing largely follows the amortizing optimization framework in Amos [2022] and other prior work, and the main part of Section 3 is mostly restating this framework in the particular setting of OT. [...] In that case, the contribution of the theoretical part of the paper reads unclear to me.
>
> I agree that the paper is mostly building on the theoretical foundations of other papers, such as Amos (2022), Taghvaei and Jalali (2019), Makkuva et al. (2020), and Korotin et al. (2019). To me, the contribution of my submission here is to connect these all together, which is shown in Remarks 3-8. I can further expand on any of these if it would be helpful, but at least to me, the connections I am making in these remarks are not evident from the original sources. I was especially surprised to realize the results in Section 3.2.2 showing that the cycle consistency term in the W2GN method of Korotin et al. (2019) can be derived from amortizing the first-order optimality conditions of the conjugate. This is novel from the perspective of W2GN, which does not take the amortization perspective, and also from Amos (2022), which does not discuss much the idea of amortizing the optimality conditions.
>
> > Though I am unsure possible or not, it may be helpful to have at least some convergence guarantee, stability analysis, or even hard instance which explicitly shows why all methods need to be sensitive to model's hyper-parameters.
>
> I am unfortunately not too sure if this is possible or not either since in the most general case, these approaches amortize the conjugate of a non-convex neural network. I have not seen much theoretical work on the convergence or stability results of amortization choices. A result in this space would very likely be significant and would impact many of the amortization settings beyond this paper too.
>
> Perhaps closer to what you were thinking of, the other interesting theoretical direction could be to characterize how a sub-optimal solution to the conjugate impacts the dual objective. This has also been elusive in the prior work, e.g., [the ICLR 2021 reviews of Korotin et al. (2019)](https://openreview.net/forum?id=bEoxzW_EXsa&noteId=yRMmEAZB2Q8) also point out that their theory assumes oracle solutions to the optimization problems. I am again unfortunately not too sure how to go about formulating the right theoretical statements or proofs necessary to make insightful experimental choices here. I will think a little more about the right way of stating this in the paper and will expand upon the “limitations” section to point the interested readers to potentially interesting open theoretical questions here.
>
> > There are many remarks in the paper, some of them seem a bit vague and in particular, require a very good knowledge on previous work to be able to understand, e.g. Remark 6, Remark 11. It may be helpful to discuss their formulations first and then distinction to this work in detail in the main text.
>
> Thank you for pointing these clarity issues out. I am not immediately sure of the best way of providing more context here, but I have noted this and will improve them for a final version of the paper.
>
> > The paper is structured nicely. The presentation and writing of the paper could be improved a little bit to facilitate readers not very familiar with the area, and save some time from memorizing all the equations and settings in different prior works while reading the paper.
>
> I agree that the paper would be difficult to read for researchers not already familiar with the related works. I originally intended the paper to be read alongside the related papers as a note on the amortization aspects behind them, but have considered adding some more context around them. Do you think it would be useful to include a lighter introduction to continuous Wasserstein-2 optimal transport in the appendix?

---

### Official Review · Reviewer_7Bdg · 2022-10-27

**Confidence:** 3
**Correctness:** 4
**Technical Novelty And Significance:** 3
**Empirical Novelty And Significance:** 3
**Recommendation:** 8

**Clarity, Quality, Novelty And Reproducibility:**

For the details of these points, please see my review of "Strength and Weakness". However, to give a high-level answer:

**Clarity:** The overall idea is very clear. Optimization terminology is sometimes vague.

**Quality:** The paper has high quality for sure. The experiments, visualizations, and results are thorough.

**Novelty:** The perspective of looking at the conjugate estimation problem as an amortized optimization problem is novel. However, I am not sure how useful it is, as there is not much theoretically appealing property rather than experimental numerical results. This is just **a** way of solving this problem.

**Reproducibility:** The paper provides all the source codes, and the main algorithm is very clear. Hence I cannot foresee any issues with reproducibility.

**Strength And Weaknesses:**

**Strength:** Firstly, I would like to thank the author for such a clear motivation, literature review, numerical experiments, implementation, and discussions. The paper is very "fun" to read, and I had a good time reading it. The focus on computing the conjugate is very essential to the relevant community.

**Weaknesses:**
*I list my major and minor concerns that hold me from being *strictly* positive about this paper. However, I still think the paper has some interesting ideas, therefore recommending an acceptance. I am also going to stay active during the discussion period in case the author has updates or questions regarding my review.*

***Major Weaknesses/Questions***
- Most of the remarks and optimization-related discussions are very redundant and they do not add much to the paper. For example, in Remark 8 it is discussed that the first-order conditions would not give a global solution for nonconvex functions. Do we even need this? I do not think any reader who wouldn't already know this will understand even page 1 of the paper.
- The tone of the author may be taken as subjective/informal or even offensive for some people. There are several **bold highlighted** sentences that are subjective and these may make the paper look like some course notes or informal discussion. Remark 7 is a good example. Also, Remark 6 "in my experiments, updating .. hurts ..." is not formal.
- Some results are not very new, but the way they are presented may sound like a breakthrough from the tone. I would like to kindly ask the author whether Section 4.1 adds anything new. Isn't this just a parallel line search implementation?
- I would be happier to see further optimization formality. Validity of $\min$ instead of $\inf$ is typically not argued. The existence of feasible solutions for, for example, duality arguments, is not discussed.
- Some problems are claimed to be "easy", for example, "**computing the exact conjugate is easy**". Is it really an easy problem? Or is that a numerical observation?
- Sometimes there are overly positive claims for simple/standard evidence. For example, in Section 3.1, the selection of a gradient for the amortization of a function is said to be well-motivated because the $\arg\min$ of the conjugate is also a derivative.
- Finally, a general question to the author: The OT setting looks relevant because we need to know the conjugate and because we would apply gradient-based learning to solve approximations of (2), but in general, this solution technique is a way of estimating the conjugate of a potential function. I am wondering, is there anything else that makes the underlying OT setting unique? Moreover, amortized optimization is already being used a lot in similar settings, may I please ask the author what makes this setting special? Apologies if I am not seeing something obvious, and again, many thanks for the work.

***Minor Weaknesses/Questions***
- Abstract has "an amortized approximations".
- Whenever $:= \arg\min$ notation is used, could the author please discuss that the $\arg\min$ set is a singleton? Otherwise, $\in \arg\min$ would perhaps be more suitable.
- "solving" the conjugation sounds a little confusing in Section 1, especially since $f$ is an optimization function. Maybe the context behind the parametrization of $f$ should be discussed first.
- Page 2: "the the exact conjugate" (typo)
- Equation (4): $f_\theta^\star(x)$ has a typo I believe.
- Equation (4): $J_{f_\theta} (x(y))$ -> this shorthand is not explained yet (shorthand for $J_{f_\theta} (x(y); y)$
- When Algorithm is first mentioned in Section 1, the "initial dual potential $f_\theta$" is not explained yet (which appears in the algorithm). Similarly, the same algorithm uses $\tilde{x}_{\varphi}(y_j)$ which is not yet discussed. Also, the same algorithm, maybe $N$ can be shown as an input, too?
- Remark 1: "$W_2$ coupling" is not abbreviated yet. Section 3.1: "MLP" not abbreviated.
- Remark 2: Perhaps the author should discuss the feasibility $x \in \mathcal{X}$, too (in order to make a weak-duality sort of argument)?
- Last paragraph of Section 2: "by predicting the solution with a model $\tilde{x}_{\varphi}(y_j)$". Solution of "what" is not clear, and $\tilde{x}_{\varphi}(y_j)$ is not introduced.
- Section 3.1, element 1: "directly maps to the solution by" -> please also state "solution of what"
- Section 3.2.1: "as optimal as possible" -> this is not a very usual terminology
- I am curious: how does the approach mentioned right after (6) relate to the standard SAA-like techniques in Stochastic Optimization?
- Page 4: ".. et al ... proposes" -> 'propose'
- Remark 7: "... state that they are not a maximin optimization procedure" -> this is not clear to me
- Question: Could the author please discuss how someone would still be interested in 3.2.3 even when the true solution to the conjugate problem is known? (for example, when we use NN based approximations for the potential?)
- Page 12, Nhan Dam et al. (2019) citation -> please capitalize "GAN".
- Conclusions: The sentence that starts with "In non-Euclidean ...." is hard to grasp.

**Summary Of The Paper:**

Monge's primal Wasserstein-$2$ transport map optimization problem admits a (Kantorovich) dual with appealing properties which enable the use of several learning/optimization algorithms to (approximately) solve the problem. However, the dual objective function includes a term involving the convex conjugate of the potential function (which is the dual optimization variable), which is hard to compute in many settings, and even impossible in some others. The author proposes taking an "amortized optimization" view for the conjugate computation problem within the iterations of estimating the potential function (e.g., while approximating it with NNs). The amortized optimization view is shown to be useful since the potential function estimation stage will involve solving the conjugate sequentially/iteratively.

**Note:** My score increased from "6" to "8" after the rebuttal.

**Summary Of The Review:**

The paper is in general written well with a very accurate literature review, the topic studied is very interesting with high relevance to both optimal transport and optimization communities. Some of the results that are discussed in detail are already well-known (if not they are easy to derive as there is no new theory, but rather a combination of known methods), the paper is written in a subjective/informal way, and in general, the paper looks like a review paper rather than a novel one. I recommend "marginally above acceptance threshold", but I do not have strong positive feelings either. I decided in favor of acceptance as the paper provides a complete set of reusable algorithms, which the community can benefit from.

---

> ### Author Response · Authors · 2022-11-18
> **Response part 1/2**
>
> Thanks very much for the thorough review! I am delighted for the positive reception and that you enjoyed reading the paper and thought it was a fun read :-)
>
> I’ve posted an updated version of the paper and here are some more responses inline to your comments and questions:
>
> > For example, in Remark 8 it is discussed that the first-order conditions would not give a global solution for nonconvex functions. Do we even need this? I do not think any reader who wouldn't already know this will understand even page 1 of the paper.
>
> I agree that remark 8 is a well-known fact in optimization but I think it is important to keep in the paper. Many existing papers such as the W2GN paper *are* using the cycle consistency term to amortize the conjugate. They do not include a discussion of the point that it optimizes the first-order optimality conditions of the conjugate and therefore could get stuck in suboptimal minima. Reviewer G2nL also brings up the fact that the meaning of this remark is not clear and I will continue thinking about the best way of presenting this detail as it does not seem to have come across with my original intention.
>
>
> > The tone of the author may be taken as subjective/informal or even offensive for some people. There are several bold highlighted sentences that are subjective and these may make the paper look like some course notes or informal discussion. Remark 7 is a good example. Also, Remark 6 "in my experiments, updating .. hurts ..." is not formal.
>
> I certainly hope that my paper does not come across as offensive and am very open to making updates to ensure this doesn’t happen. I have added a few other clarifications on these, and please let me know if you have any other concrete suggestions here. Sometimes these stylistic details are hard for me to decide on, especially when people perceive my stylistic choices differently, e.g., in contrast, Reviewer G2nL states that “*the clarity of the paper is very good and sufficient code snippets and algorithm boxes were provided for reproducibility. I personally like the style of using bold text*”.
>
> In Remark 7, I am expressing disagreement with methods that focus on the formulation as a maximin optimization problem. Here, I explicitly state what I disagree with (*methods such as W2GN distinguishing themselves from methods such as Makkuva et al. by their formulation as a maximin optimization problem*), why I disagree with it (*because it is a red herring and both methods still inaccurately approximate the conjugate*), and what I propose instead (*to focus on the amortization differences*). I believe this is a subtle but important point and hope I have not expressed it in a way that would be offensive to anybody. I am open to suggestions for improving this.
>
> And, I have removed the word “hurts” from remark 6.
>
> > Some results are not very new, but the way they are presented may sound like a breakthrough from the tone. I would like to kindly ask the author whether Section 4.1 adds anything new.  Isn't this just a parallel line search implementation?
>
> I have moved all of the parallel line search details (what was previously section 4.1) to the appendix and have toned down the writing everywhere. Please let me know if there’s anything else on the presentation of the line search details that I can improve!
>
> > I would be happier to see further optimization formality
>
> I have tried to add more justification around the min and inf operations and to existence results, e.g. Santambrogio (2015, Theorem 1.17) and the settings of Taghvaei and Jalali (2019) and Makkuva et al. (2020).

---

> > ### Author Response · Authors · 2022-11-18
> > **Response part 2/2**
> >
> >
> > > Some problems are claimed to be "easy", for example, "computing the exact conjugate is easy". Is it really an easy problem? Or is that a numerical observation?
> >
> > I have tried to qualify “easy” everywhere I use it, e.g. in the quoted text, by saying it is “computationally easy”. I agree “easy” is somewhat of a vague term but other works (like the one I quoted in the beginning) state that the conjugation is “hard”, so I was at least trying to say that conjugation isn’t as hard as the other works make it out to be. I am open to better-qualifying and clarifying what I mean by “easy” if it is still ambiguous anywhere.
> >
> > > Sometimes there are overly positive claims for simple/standard evidence. For example, in Section 3.1, the selection of a gradient for the amortization of a function is said to be well-motivated because the argmin of the conjugate is also a derivative.
> >
> > I think it is well-motivated to use the derivative of a function to approximate the argmin of the conjugate because the derivative of the conjugate is the argmin. I think it’s an interesting point for somebody thinking about how to instantiate an amortization model because otherwise taking the gradient of a function to be the amortization model instead of just an MLP in general is somewhat counter-intuitive. I am open to modifying this if you are suggesting that I replace the word “well-motivated” in the text with something else.
> >
> > >  I am wondering, is there anything else that makes the underlying OT setting unique? Moreover, amortized optimization is already being used a lot in similar settings, may I please ask the author what makes this setting special? Apologies if I am not seeing something obvious, and again, many thanks for the work.
> >
> > I agree! The idea of amortizing and fine-tuning conjugates is applicable far beyond the optimal transport setting here and in theory could be a nice computational tool anywhere people are interested in computationally approximating the conjugate. In fact, I have noticed [another ICLR submission](https://openreview.net/pdf?id=c9lAOPvQHS) that is amortizing the convex conjugate to compute the inverse Fisher for the natural gradient: in their notation the amortization model is $Q(\lambda)$ and it is learned in their equation (15) to amortize the objective of a conjugate, and is then used in equation (17) to update the model’s loss with an approximate natural gradient.
> >
> > > Minor Weaknesses/Questions
> >
> > Thanks also for all of the corrections on these details. I think I have clarified most of them in the paper and will go through again for a final version and make sure I’ve addressed everything. On equation (4), I am not sure what typo you saw in it, and I also defined $J$ above in equation (3) — I added a pointer here to clarify it.
> >
> > > I am curious: how does the approach mentioned right after (6) relate to the standard SAA-like techniques in Stochastic Optimization?
> >
> > I am not too sure, do you have a more specific reference in mind for this?
> >
> > > Question: Could the author please discuss how someone would still be interested in 3.2.3 even when the true solution to the conjugate problem is known? (for example, when we use NN based approximations for the potential?)
> >
> > Is your question on why regressing onto the true solution is useful when the true solution is already available? If so, it’s for computational reasons. The true solution may take many iterations to obtain when solved from scratch, e.g. as Figure 2 shown for L-BFGS, but can be obtained to the same level of precision when starting from an amortized prediction. In Algorithm 1, I propose to interweave the solving and amortization: the solver always starts with the amortized prediction, which hopefully converges in a few iterations, and the amortization model is regressed onto the improved solution so that hopefully the starting point is even better in the next solve.

---

> > > ### Comment · Reviewer_7Bdg · 2022-12-03
> > > **Acknowledging the rebuttal**
> > >
> > > Dear Author,
> > >
> > > Many thanks for your answers. Apologies for my late reply. The rebuttal came late in the discussion deadline and I thought I cannot reply anymore -- but I can see that I am still allowed to post replies.
> > >
> > > I would like to express my thanks for answering most of my questions and addressing the others in the paper. The answer to your question "Is your question on why..." is affirmative, and your response is great.
> > >
> > > There are some parts where I do not agree, but these are about writing style and do not change anything about the value of the paper and my review: I would not prefer a paper where the author says words like "I disagree", or highlight simple things like (previously) Remark 8.
> > >
> > > Overall, I like the paper. I am increasing my score by 1 point because of the author's responses. I think the proposed approach will be adopted in more general ML settings, which is exciting.
> > >
> > > Best regards.

---

> > > > ### Author Response · Authors · 2022-12-03
> > > > **Response**
> > > >
> > > > Thank you! That is all great to hear, and I will continue thinking about the best ways of presenting remarks 7 and 8.

---

### Official Review · Reviewer_eQGj · 2022-10-27

**Confidence:** 2
**Correctness:** 4
**Technical Novelty And Significance:** 2
**Empirical Novelty And Significance:** 3
**Recommendation:** 6

**Clarity, Quality, Novelty And Reproducibility:**

The paper is mostly *clear*, although, as mentioned above, some parts of the text are unclear. (An additional point: eq. (2) is missing a square in the RHS.)

*Reproducibility* is partial/laborious: the appendix contains written code for the examples, but, to my knowledge, there's no way to download and run code directly.

As far as I can tell, the present paper is *novel*: the first to try and compute conjugate maps for optimal transport via "fast methods".

*Quality*: the main evidence that the method is good is the experiments presented, which are adequate. However, I find them a bit incomplete, in that I'd like to see a "real task" where OT is typically used and the present method leads to improvements.  Also, there's the tuning issue mentioned above.


**Strength And Weaknesses:**

*Strenghts*

- The experimental results look promising.
- The paralellized Armijo is a cool idea.

*Weaknesses*

- I found the writing a bit difficult for a non-expert (eg. terms like "amortization" are never defined, and is only really explained in the code in the appendix).
- There's no theory to back up the methods.
- Some parts of notation are not well defined (eg. the letter for the dimension changes from n to D in the text).
- It is not clear to me how the method should be tuned in the abscence of a ground truth.


**Summary Of The Paper:**

The results in this paper apply to the computation of optimal transport maps for the Wasserstein-2 distance in Euclidean space. Under weak assumptions, these maps are given by gradients of convex functions. Moreover, the convex function and its Fenchel-Légendre conjungation are a solution to the dual problem. The paper presents different frameworks for performing the conjugation via a differentiable optimization problem. Moreover, the author also provides an efficient and paralelizable "Armijo-type" algorithm for performing the conjugation. Experiments suggest that this approach leads to significant improvements in the computation of optimal transport maps.



**Summary Of The Review:**

The paper presents novel methods for the conjugation step in dual Wasserstein-2 optimal transport. Experiments suggest that the methods are good, but the paper is a bit hard to read, and leaves some questions regarding tuning unresolved.

---

> ### Author Response · Authors · 2022-11-18
> **Response**
>
> Thanks for the review! I agree that the paper would be difficult to read for researchers not already familiar with the related works. I originally intended the paper to be read alongside the related papers as a note on the amortization aspects behind them, but have considered adding some more context around them. Do you think it would be useful to include a lighter introduction to continuous Wasserstein-2 optimal transport in the appendix?
>
> I’ve posted an updated version of the paper and here are some more responses inline to your comments and questions:
>
> > I found the writing a bit difficult for a non-expert (eg. terms like "amortization" are never defined, and is only really explained in the code in the appendix).
>
> I’ve tried to define “amortization” in the text by stating it’s meant to approximate the solution to optimization problems. I just went through and tweaked the language a little more. I’ve updated the abstract to read:
>
> *To overcome this, the computation of the conjugate can be
> approximated with amortized optimization, which learns a
> model to predict the conjugate.*
>
> And the beginning of section 3 to be:
>
> *This is an instance of \emph{amortized optimization} methods
> which predict the solution to a family of
> optimization problems that are repeatedly solved*
>
> > There's no theory to back up the methods.
>
> While my paper does not present any new theoretical results, the methods are theoretically rooted in prior work on optimal transport and learning continuous potentials, especially on Taghvaei and Jalali (2019), Makkuva et al. (2020), and Korotin et al. (2019). None of these papers have any theory on how the approximation quality of the conjugate impacts the dual learning, e.g., [the ICLR 2021 reviews of Korotin et al. (2019)](https://openreview.net/forum?id=bEoxzW_EXsa&noteId=yRMmEAZB2Q8) also point out that their theory assumes oracle solutions to the optimization problems. My paper scopes only to the conjugate computation part of these methods and shows how to 1) view prior approaches as approximate amortizations of the conjugate, and 2) better-approximate the conjugate by combining amortization and fine-tuning.
>
> > Some parts of notation are not well defined (eg. the letter for the dimension changes from n to D in the text).
>
> I originally used $D$ to present the results on the Wasserstein-2 benchmark to stay consistent with their notation but I now see how it is confusing. I have updated the paper to use $n$ everywhere. Please let me know if there are any other ill-defined pieces of notation
>
> > It is not clear to me how the method should be tuned in the abscence of a ground truth.
>
> What ground truth are you referring to here? Is it the ground-truth solution of the conjugate, or the ground-truth optimal transport map between the measures? If it is either of those, it is possible to learn the optimal potentials and conjugates without those.
>
> > An additional point: eq. (2) is missing a square in the RHS.
>
> Eq (2) is the dual form directly from Villani eq (5.12) and it does not need to be squared
>
> > but, to my knowledge, there's no way to download and run code directly.
>
> I have attached an anonymized version of the code to the supplementary material here that I will open source along with the paper. The code reproduces every figure, table, and result in the paper and there is a README in there with more instructions on running them.
>
> >  Quality: the main evidence that the method is good is the experiments presented, which are adequate. However, I find them a bit incomplete, in that I'd like to see a "real task" where OT is typically used and the present method leads to improvements.
>
> I agree! Almost all of the other methods for continuous OT include a component amortizing the computation of the conjugate and I would love to evaluate improved conjugates in their settings. However I unfortunately am not aware of any settings as standardized and widely available as the NeurIPS 2021 Wasserstein benchmark, which is why I chose to focus on that first. That benchmark contains an evaluation of the best methods for computing continuous OT maps and my results show that better-computing the conjugate can significantly improve the $\mathcal{L}^2$-UVP quality of the learned transport maps by ~200-400%.

---

> > ### Comment · Reviewer_eQGj · 2022-11-30
> > **Thanks!**
> >
> > Thank you for your detailed response. To answer your question:
> >
> > "Do you think it would be useful to include a lighter introduction to continuous Wasserstein-2 optimal transport in the appendix?"
> >
> > If you can do this in a way that helps the reader understand the notation, I am all for it.

---

### Decision · Program_Chairs · 2023-01-20

**Decision:**

Accept: poster

**Justification For Why Not Higher Score:**

see meta-review - this is an interesting submission, but with a somewhat narrow scope - still a very 'cool' idea, as written by one of the reviewers.

**Justification For Why Not Lower Score:**

see meta-review

**Metareview: Summary, Strengths And Weaknesses:**

This work proposes a novel method for computation of convex conjugates in optimal transport problems with Euclidean costs.

The reviewers and I agree that this is a very good submission that presents new, interesting ideas, in a well-exposed manner.

**Note From Pc:**

if the above contains the word "oral" or "spotlight" please see: "oral" presentation means -> notable-top-5% and "spotlight" means -> notable-top-25%. As stated in our emails, we are disassociating presentation type from AC recommendations